# Global unleashing of transcription elongation waves in response to genotoxic stress restricts somatic mutation rate

Matthieu D. Lavigne [1], Dimitris Konstantopoulos[1], Katerina Z. Ntakou-Zamplara[1], Anastasios Liakos[1] & Maria Fousteri [1]

Complex molecular responses preserve gene expression accuracy and genome integrity in the face of environmental perturbations. Here we report that, in response to UV irradiation, RNA polymerase II (RNAPII) molecules are dynamically and synchronously released from promoter-proximal regions into elongation to promote uniform and accelerated surveillance of the whole transcribed genome. The maximised influx of de novo released RNAPII correlates with increased damage-sensing, as confirmed by RNAPII progressive accumulation at dipyrimidine sites and by the average slow-down of elongation rates in gene bodies. In turn, this transcription elongation 'safe' mode guarantees efficient DNA repair regardless of damage location, gene size and transcription level. Accordingly, we detect low and homogenous rates of mutational signatures associated with UV exposure or cigarette smoke across all active genes. Our study reveals a novel advantage for transcription regulation at the promoter-proximal level and provides unanticipated insights into how active transcription shapes the mutagenic landscape of cancer genomes.

[1] Biomedical Sciences Research Center 'Alexander Fleming', 34 Fleming Street, Vari, 16672 Athens, Greece. Dimitris Konstantopoulos and Katerina Z. Ntakou-Zamplara contributed equally to this work. Correspondence and requests for materials should be addressed to M.F. (email: fousteri@fleming.gr)

Environmental stress damages DNA and perturbs transcription and replication, thus impaired or overwhelmed DNA damage responses (DDR) will limit cell homeostasis recovery and promote mutagenesis, uncontrolled cell proliferation or cell death, leading to tissue dysfunctions, disease pathogenesis and ageing[1–8]. Recent mapping of ultraviolet (UV) irradiation- and smoking-associated cancer genomes revealed an intriguing heterogeneity in the distribution of somatic mutations that implies a central role of transcription in maintaining DNA

integrity[8–16]. While transcriptionally inactive regions solely rely on Global Genome-Nucleotide Excision Repair (GG-NER), the sensing activity of actively elongating RNA polymerase II (RNAPII) on interfering DNA lesions, caused by exposure to UV, tobacco smoke and other chemicals such as cisplatin, yields faster repair kinetics in the transcribed genome via the targeted action of transcription-coupled NER pathway (TC-NER)[8,15–18]. Therefore, ongoing transcription safeguards genome stability[13,15,16] and can contribute to guaranteeing low mutation frequencies in

**Fig. 1** Global triggering of transcription waves on virtually all active genes in response to UV irradiation. **a** Heatmaps illustrating the $Log_2$ fold change (FC) of main RNAPII isoforms comparing reads density (Rd) between UV irradiation (8 J m$^{-2}$) and steady-state (FC = (Rd$_{+UV}$)/(Rd$_{NO\ UV}$), as aligned at individual (rows) regions (−250 bp to +2 kb relative to TSS) and categorised by gene expression status (see Supplementary Fig. 1). **b** RNAPII Escape Indexes from promoter regions (EI) before and after UV plotted as empirical cumulative distribution of fraction (ECDF) of active (labelled in solid green throughout the paper) genes. EI represents the ratio of reads density in gene body over reads density in promoter. **c** Individual comparison of EI before and after UV for all active and poised genes. Percentages of genes with increased escape after UV (ΔEI > 1, dark green dots) are shown. Chi-square test ($\chi^2$) determines whether observed number of genes with ΔEI > 1 differs from expected value purely by chance. P is indicated. **d** Heatmaps illustrating RNAPII-ser2P and Input read densities from TSS to TSS + 60 kb before and after UV, as aligned at individual (rows) active genes larger than 60 kb and ranked by increasing EI (as determined before UV in **b**). **e** Average plots of read densities shown in **d** indicating progression of wave front position (kb) at arbitrary threshold representing the transition state (enriched to steady-state). **f** Correlation plot between constitutive EI (NO UV) for RNAPII-ser2P (left) or RNAPII-ser5P (right) and EI change after UV (ΔEI(+UV vs NO UV)) for all active genes. PCC scores are indicated. **g** Same as in **f** for RNAPII-ser2P ΔEI (+UV vs NO UV) plotted against gene length (bp)

the transcribed strand of active genes[7,8,10–12,14]. Nonetheless, transcription-dependent efficiency in DNA surveillance contrasts with prevailing views, which point to DNA damage-induced inhibition of transcription[19,20].

Productive transcription elongation is regulated by the accurate and coordinated release of RNAPII from promoter-proximal pause (PPP) sites located immediately (~60 bp) downstream of transcription start sites (TSS)[21,22] and occurs in defined sets of genes in response to various developmental[23,24] or environmental cues[25–27]. Mechanistically, positive transcription elongation factor b (P-TEFb) complex is recruited at PPP sites and is licensed by the activation of the Cdk9 kinase module[28,29]. This switch enables phosphorylation of (i) the negative elongation factor (NELF), which is removed from PPP sites, (ii) the DRB sensitivity-inducing factor (DSIF), which is transformed into a positive elongation factor, and (iii) the serine 2 (Ser2) residues of the heptapeptide repeats of the C-terminal domain (CTD) of RNAPII main subunit rpb1[30–33]. As a result, waves of transcription elongation sweep along the gene[34]. It has been reported that UV irradiation activates calmodulin–protein phosphatase 2B and 1A (PP2B and PP1A) signalling pathways and provokes the disruption of 7SK snRNP inactive complex, a major reservoir of poised P-TEFb in the nucleus, leading to a rapid increase in the pool of active P-TEFb[29,35]. Nevertheless, no further studies that address the transcriptional relevance of P-TEFb activation in response to UV have been reported.

Herein, we identified a novel metabolic function associated with active transcription elongation that is triggered immediately upon UV irradiation. Global de novo stress-induced RNAPII elongation waves are simultaneously released from promoter-proximal pausing sites, in virtually all expressed genes, regardless of gene size or expression level. Thorough comparisons of our functional genomics data with previously published data sets[7,36] further demonstrated that the identified transcription response has critical impact on the accelerated and unbiased removal of damaged DNA caused by environmental challenges. A striking consequence is the homogenous decrease of the mutation burden throughout the transcribed genome in UV- and smoke-associated cancers. Our findings therefore substantiate the importance for cells to pre-wire for prompt release of RNAPII into elongation. In addition, our data reveal unanticipated functional and molecular insights into an advanced transcription-dependent defence mechanism that transiently adjusts the regular transcription-firing program of active genes in human cells to accelerate damage-sensing and to actively preserve genetic integrity.

## Results

**UV stress triggers RNAPII elongation waves in active genes.** To obtain a precise and genome-wide view into the molecular events controlling transcription reorganisation in response to genotoxic stress, we generated high-throughput profiles of RNAPII occupancy (chromatin immunoprecipitation-sequencing (ChIP-seq)) and ongoing transcription activity (nascent RNA-seq) in hTERT immortalised normal human foreskin fibroblasts (see Methods). We took advantage of previously established protocols to synchronise cells in G1 to limit cell-cycle heterogeneity and to achieve steady-state levels of RNAPII and nascent transcripts across the transcribed region (gene body). Briefly, serum-starvation for 72 h at confluency enriched for cells in G0/G1 (Supplementary Fig. 1a) and after release in complete medium for 3 h prior to UV irradiation (see Methods) we allowed for the rapid recovery of steady-state levels of transcription activity[37] (Supplementary Fig. 1b). Next, we determined gene activity status (active, poised or inactive) by intersecting RNAPII Chip-seq

peaks around promoters for each of the commonly described isoforms of RNAPII CTD (pre-initiating (RNAPII-hypo), initiating (RNAPII-ser5P, from TSS to PPP) and elongating (RNAPII-ser2P, after PPP)) (Supplementary Fig. 1c, d). We also confirmed that the levels of nascent RNA (nRNA) and mRNA at steady state were correlated with the RNAPII occupancy status (Supplementary Fig. 1e–g).

Stress, triggered by low doses (8 J m$^{-2}$) of UV-C irradiation, caused an extensive reorganisation of transcription. Particularly in active genes, RNAPII isoforms displayed important changes in their occupancy profiles (Fig. 1a and Supplementary Fig. 2a, b). Unexpectedly, we detected a substantial increase of elongating polymerases in the gene bodies after the TSS and a concomitant depletion of promoter-bound RNAPII (−250 to +100 bp from TSS), especially for the hypophosphorylated form of RNAPII (RNAPII-hypo), as anticipated[38]. To acquire a more quantitative view, we calculated the ratio of RNAPII reads between gene body and promoter regions (Escape Index: EI; see Methods) before and after DNA damage induction for each gene and for all isoforms of RNAPII. We found a transient yet significant increase of EIs ($\Delta EI > 1$) at early time points during recovery (0.5 to 2 h post UV) at the vast majority (up to 90.6 %) of active genes for RNAPII-ser2P and for RNAPII-ser5P (Fig. 1b, c, and Supplementary Fig. 2c, d). In contrast, poised and inactive genes showed fewer alterations in escape ratios (Fig. 1b, c and Supplementary Fig. 2c, d) suggesting that these genes, which are not normally transcribed, are less affected by the aforementioned transcription reorganisation.

Our isoform-specific ChIP-seq analyses clarified the general views offered by total RNAPII binding profiles[38], as we uncoupled each fundamental transcription stage. We hypothesised that the increase in EI was due to the switch of promoter-proximal paused RNAPII to productive elongation[22]. Consistent with this assumption, we found that there was a concomitant decrease in RNAPII-ser5P and RNAPII-Ser2P read densities at promoter-proximal regions (−250 to +100 bp around TSS, Supplementary Figs. 2b and 3a–d) and a gain in read density in gene bodies (+100 bp to 2000 bp after TSS), especially for RNAPII-ser2P (more significant shift of RNAPII-ser2P reads into gene bodies) (Supplementary Figs. 2b and 3e–g). On the contrary, RNAPII-hypo EI was affected after UV only because of the very significant loss of reads at promoters and not because of changes in read density in the gene body (Supplementary Figs. 2b and 3a–d). Overall, this RNAPII redistribution around promoter regions supported a widespread release of transcription elongation waves progressing with gradually decreasing rates from the 5′ to the 3′ end of genes (Fig. 1d, e and Supplementary Fig. 3h, i, see below).

We also analysed the changes in RNAPII isoforms bound to chromatin by western blot analysis of inputs samples (Supplementary Fig. 3j) and of total RNAPII-enriched material (anti-rpb1, Supplementary Fig. 3k). As expected, the levels of pre-initiating polymerases (RNAPII-hypo) were significantly reduced by 75% 2 h post UV (log$_2$ FC ~ −2), thus matching the changes in levels estimated by RNAPII-hypo ChIP-seq (Supplementary Fig. 3c) and further extending previously published biochemical evidence[19,39]. In contrast, we did not find such extensive changes in the levels of initiating and elongating complexes (RNAPII-ser5P and -ser2P, respectively) during the early recovery phase (+0.5 to +2 h) (Supplementary Fig. 3). We thus conclude that, during the early stages of stress recovery the influx of actively elongating RNAPII-ser2P molecules in gene bodies (Fig. 1c–e) is fed by the transformation of pre-initiation complexes (PIC, RNAPII-hypo) into initiating (ser5P) complexes, which are promptly released into productive elongation (ser2P).

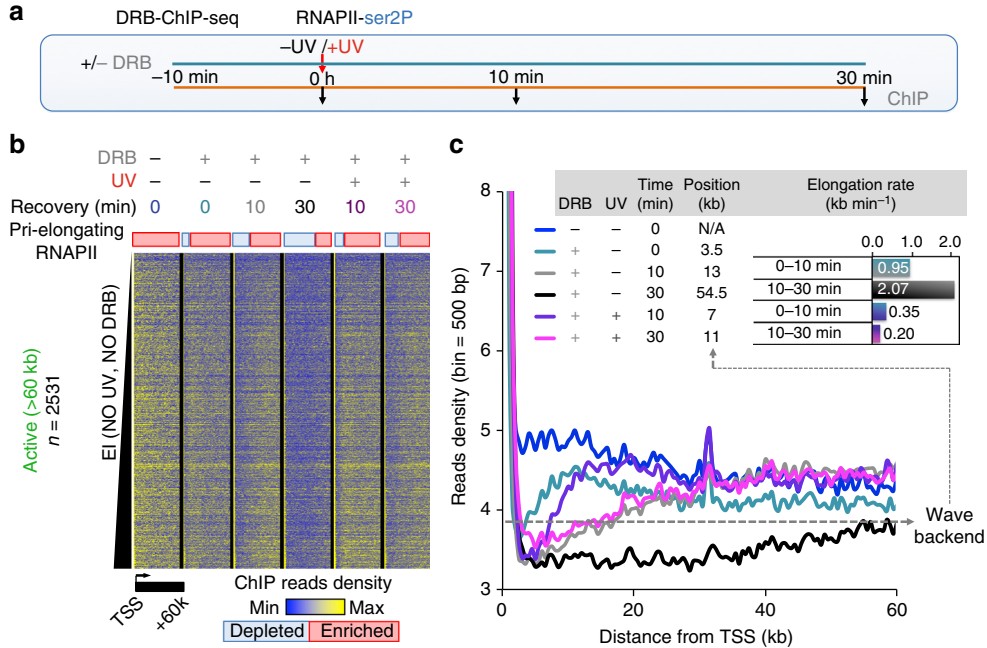

**Fig. 2** Inhibition of RNAPII transition into elongation unmasks the kinetics of RNAPII molecules elongating prior to UV stress (pri-elongating). **a** Scheme of DRB-inhibition methodology (DRB-ChIP-seq), UV irradiation was performed at 20 J m$^{-2}$, see also Methods. **b** Heatmaps illustrating RNAPII-ser2P read densities from TSS to TSS + 60 kb, for samples defined in **a**, as aligned at individual (rows) active genes larger than 60 kb and ranked by increasing steady-state EI (see Fig. 1d). **c** Average plots of read densities for RNAPII-ser2P derived from **b** highlighting the pri-elongating RNAPII wave backend positions at arbitrary threshold (dashed line) representing the transition state (depleted to enriched for RNAPII-ser2P). Insert shows corresponding average (n = 2531) elongation rates calculated from differences between wave backend positions in the considered time interval. Note: gradients of colours reflect the transition between indicative time points

Accordingly, we found that regions directly downstream of PPP do display substantial increases in nRNA levels between 0.5 and 2 h post UV (Supplementary Fig. 4a). Nonetheless, there was an overall reduction of RNA synthesis caused by the significant depletion of reads in more distal regions of the gene bodies, thus underlying the global decrease of RNA production in the nuclei 2 h after UV irradiation (Supplementary Fig. 4a–c). This was correlated with the fact that transcription waves tend to slow-down as they travel along the genes (see Supplementary Fig. 3i), probably because of the increased proportion of elongating RNAPII encountering DNA lesions. Finally, at late recovery time points, we observed a reset of steady-state levels of RNAPII genome-wide distributions, EIs and nascent RNA levels thus suggesting that efficient DNA repair drives transcription recovery (see Fig. 1d, e and Supplementary Figs. 2d, 3h, j, k, and 4a, c). Together these data update the views on the previously described transient UV-induced inhibition of RNA synthesis[19,20,40,41] and help to resolve the paradox on how cells can take advantage of TC-NER[16] pathway despite the apparently low transcriptional activity (see below).

This widespread release and propagation of waves of productively elongating RNAPII travelling across thousands of active genes (synchronised waves) affects the largest single group of co-regulated target genes for which RNAPII is released from PPP sites in response to a specific stimuli[26,42,43]. Exposure to potentially mutagenic agents that threaten DNA integrity may temporarily force the cells to bypass the normally strictly regulated transcription checkpoint defined by PPP. Accordingly, we demonstrate that EIs of normally highly paused/low-escaping genes are more significantly increased than EIs of less paused (highly expressed) genes (anti-correlation, Fig. 1f). In other words, RNAPII levels in gene bodies after UV differ from the steady-state levels (more uniform yellow signal in Fig. 1d upon UV) and losses of early-elongating RNAPII reads around PPP sites are more pronounced for less expressed genes (Supplementary Fig. 3c). Given that expression levels are reasonably correlated to EI (Pearson's correlation coefficient (PCC) = 0.3711) and that entry of RNAPII into gene bodies is also not correlated to gene size (Fig. 1g), we conclude that, upon stress, this mechanism enables prompt release of RNAPII molecules from PPP regions even at normally less frequently transcribed genes providing important functional advantages to the cell (see below).

**De novo release of RNAPII elongation waves.** To confirm that the release of RNAPII from PPP sites occurs de novo, we blocked the transition of RNAPII into productive elongation[22] with 5,6-dichloro-1-β-D-ribofuranosylbenzimidazole (DRB), a widely used inhibitor than can reversibly prevent RNAPII PPP-release (Fig. 2a). DRB treatment during UV irradiation and recovery was sufficient to cancel the aforementioned stress-dependent RNAPII wave generation and propagation in virtually all active genes (Supplementary Fig. 5a), as confirmed by the dramatic decrease in RNAPII escape from promoter-proximal regions (Supplementary Fig. 5b). This treatment also allowed us to uncouple the dynamics of two previously indistinguishable subclasses of elongating RNAPII molecules: the ones that are already engaged in elongation prior to stress (pri-elongating) and the de novo PPP-released polymerases. Focusing on pri-elongating RNAPII molecules profiles, we found that there was a substantial retain of RNAPII ChIP-seq reads upon UV in the distal parts of long genes (Fig. 2b, c: compare black and pink). This result was corroborated by the concomitant detection of significant levels of nascent RNA in these regions in active genes even up to 2 h post UV (50–100 kb, see Supplementary Fig. 4b). Notably, meta-analysis of previously reported nascent RNA data[44] mirrored our findings (see Supplementary Fig. 4b).

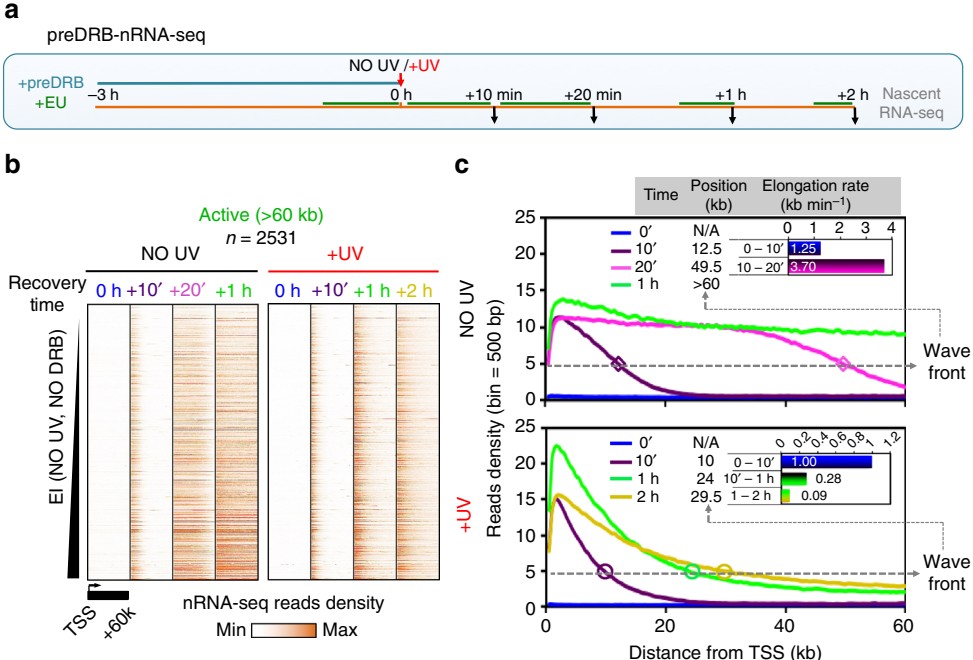

**Fig. 3** Stress-triggered de novo transcription influx decelerates progressively. **a** Scheme of DRB-release methodology (preDRB-nRNA-seq), UV irradiation was performed at 20 J m$^{-2}$ see also Methods. **b** Heatmaps illustrating nascent RNA read densities from TSS to TSS + 60 kb, for samples defined in **a**, as aligned at individual (rows) active genes larger than 60 kb and ranked by increasing steady-state EI (Fig. 1d). **c** Average plots of read densities for nascent RNA derived from **b**, highlighting de novo wave-release of RNAPII. Differences in wave front positions at an arbitrary threshold (dashed line representing the transition from enriched levels to steady-state levels) were used to calculate average (n = 2531) elongation rates in the indicated time intervals (inserts)

Therefore, although elongation rates of pri-elongating complexes were decreased (Fig. 2c and Supplementary Fig. 5c, d), our data do not support the idea of a total loss of ongoing elongation as suggested previously[44]. We thus conclude that a fraction of RNA polymerases already engaged on gene bodies at the time of stress continues elongating even in distal loci, regardless of the distance to TSS (see below).

We analysed independently de novo PPP-released RNAPII molecules and measured nascent RNA synthesis in cells that had been incubated with DRB prior to UV irradiation (Fig. 3a, see also Methods). This treatment led to elimination of elongating RNAPII and nascent RNA reads from gene bodies at the time of stress (Fig. 3b, c; time = 0 h). Following the removal of the DRB-mediated elongation block, non-irradiated cells quickly resumed normal transcription patterns (Fig. 3b, c). Surprisingly, but in accord with a very recent report[45], UV irradiation did not abolish the recovery of nascent transcription after DRB removal (Fig. 3b, c). In fact, despite DRB pre-treatment, UV did trigger a wave-release of RNAPII in essentially all active genes, similarly to non-DRB treatment (Supplementary Fig. 6 and see Fig. 1e). Therefore, we suggest that UV irradiation is by no means sufficient to prevent release of RNAPII into productive elongation.

Transcription elongation rates of the de novo released RNAPII complexes decreased during recovery, in a similar way to the pri-elongating ones (Figs. 2c and 3c), confirming the changes measured overall ($R^2 = 0.89$, compare Fig. 3c with Supplementary Fig. 3i). Nascent RNA production was also found to be lower, but not absent, in more distal parts of gene bodies (see Fig. 3c and Supplementary Fig. 4a). We thus conclude that, de novo released RNA polymerases also progressed eventually through the genes, albeit slowly. In agreement, even at later time points during recovery, RNAPII wave front proceeded incrementally between +2 h and +6 h (Fig. 1e).

We, therefore, propose the following scenario in response to genotoxic threat: first, there is a significant increase in nascent RNA levels in the proximal part of active genes, because more RNAPII molecules switch to a productive elongation state throughout the active gene set. Next, synthesis rate is rapidly and gradually affected along gene bodies (towards more distal regions), and we anticipate that DNA damages slow-down both de novo released and already transcribing RNAPII (see below). Importantly, this model is consistent with and explains the recent findings suggesting that transcription initiation and elongation still occurs in proximal areas upon UV-irradiation although progress into gene bodies is very slow[45].

**De novo released transcription enhances DNA damage-sensing.** Understanding the advantage of a combined widespread release of de novo elongating complexes gradually decelerating in gene bodies could answer an important open question: why cells need to prime such an extended proportion of active genes for potential prompt pause-release[22,23]. Considering the increased probability that a progressing RNAPII will encounter a lesion in a travelling distance- and time-dependent manner, we anticipate that as more actively elongating RNAPII molecules are released into gene bodies, more frequent and faster transcription-dependent DNA repair will occur.

First, we checked whether there is a functional link between the probability of RNAPII stalling and the detection of DNA lesions. A higher UV dose (20 J m$^{-2}$), which induces more UV lesions, produced an equally widespread release of RNAPII in virtually all active genes (Supplementary Fig. 7a). On the other hand, the greater number of lesions was sufficient to significantly increase the number of RNAPII molecules that may stall at lesions at a given time point, as shown by further impairments in the average progression of the RNAPII wave (Supplementary Fig. 7).

More direct evidence of the role of RNAPII in detecting lesions was revealed by the time-dependent increase in the accumulation

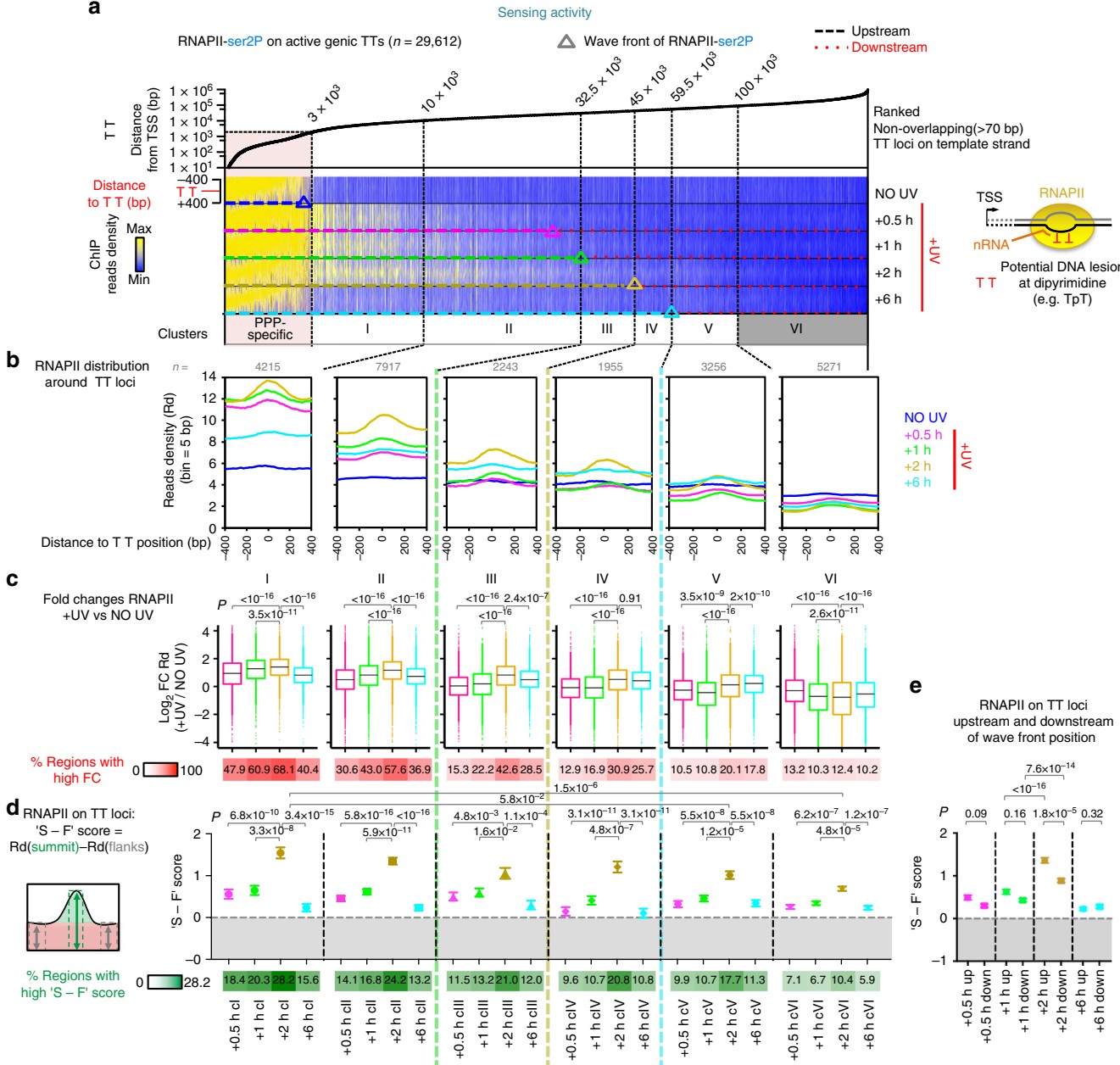

**Fig. 4** RNAPII-dependent sensing of DNA lesions is enhanced by the de novo wave-release. **a** Heatmaps illustrating the distribution of RNAPII-ser2P reads aligned around TT loci localized in active genes (as defined in Supplementary Fig. 8a) before (NO UV) and after UV irradiation (+UV, 8 J m$^{-2}$) and sorted from left to right by increasing distance relative to TSS. TT loci were clustered (upstream (i.e. Clusters I, II, III for +2 h) or downstream (i.e. Clusters IV, V, VI for +2 h)) relatively to RNAPII-ser2P wave front positions, which indicate the transition state from de novo-enriched to pri-elongating only population of RNAPII (see Fig. 1e, Supplementary Fig. 8d and Methods for details). TT loci near PPP-specific RNAPII signal were not considered for analysis. **b** Average plots of read densities (Rd) as mapped in **a** for individual clusters (n is indicated). **c** Top; box plots showing the changes in Rd in regions analysed in **b** and normalised by Rd of NO UV (P value based on two-sided t-test using the Benjaminin–Hoechberg (BH) adjustment). Boxes refer to the first quartile, the median and the third quartile. Whiskers refer to the 10–90% interquantile range. Bottom; heatmap representing the proportion of regions (per cluster) with high fold changes (FC > 2) in Rd. **d** Top; plot showing the average (± SEM) of the difference between Rd at summit and Rd at flanks (defined as 'S - F' score) of all regions analysed in **b** (P-value based on two-sided Wilcoxon rank-sum test the BH adjustment). Bottom; heatmap representing the proportion of regions (per cluster) with high 'S - F' scores (> threshold = average[S - F]$_{exon\ start}$ + 3×SD[S - F]$_{exon\ start}$, see Supplementary Fig. 9 and Methods). **e** Plot showing the comparison of average (± SEM) 'S - F' score for all regions upstream (Up) and downstream (Down) of the respective wave front position, for each time of recovery (P-value based on two-sided Wilcoxon rank-sum test using BH adjustment)

of RNA polymerases at intragenic sites of potential DNA adducts (i.e. TpT dipyrimidine (TT), Supplementary Fig. 8a, b). We based our analysis on the known facts that TT dinucleotides loci are the most frequently dimerised pyrimidines after UV exposure[46] and that damage profiles of UV-induced cyclobutane pyrimidine

dimers (CPDs) are dictated by TT frequency[47]. We gained insights into the dynamics of DNA lesion recognition by RNAPII, as we sorted the TT sites in active genes according to their distance to TSS. We then clustered TT loci by their relative position to RNAPII wave fronts during recovery (Fig. 4a and

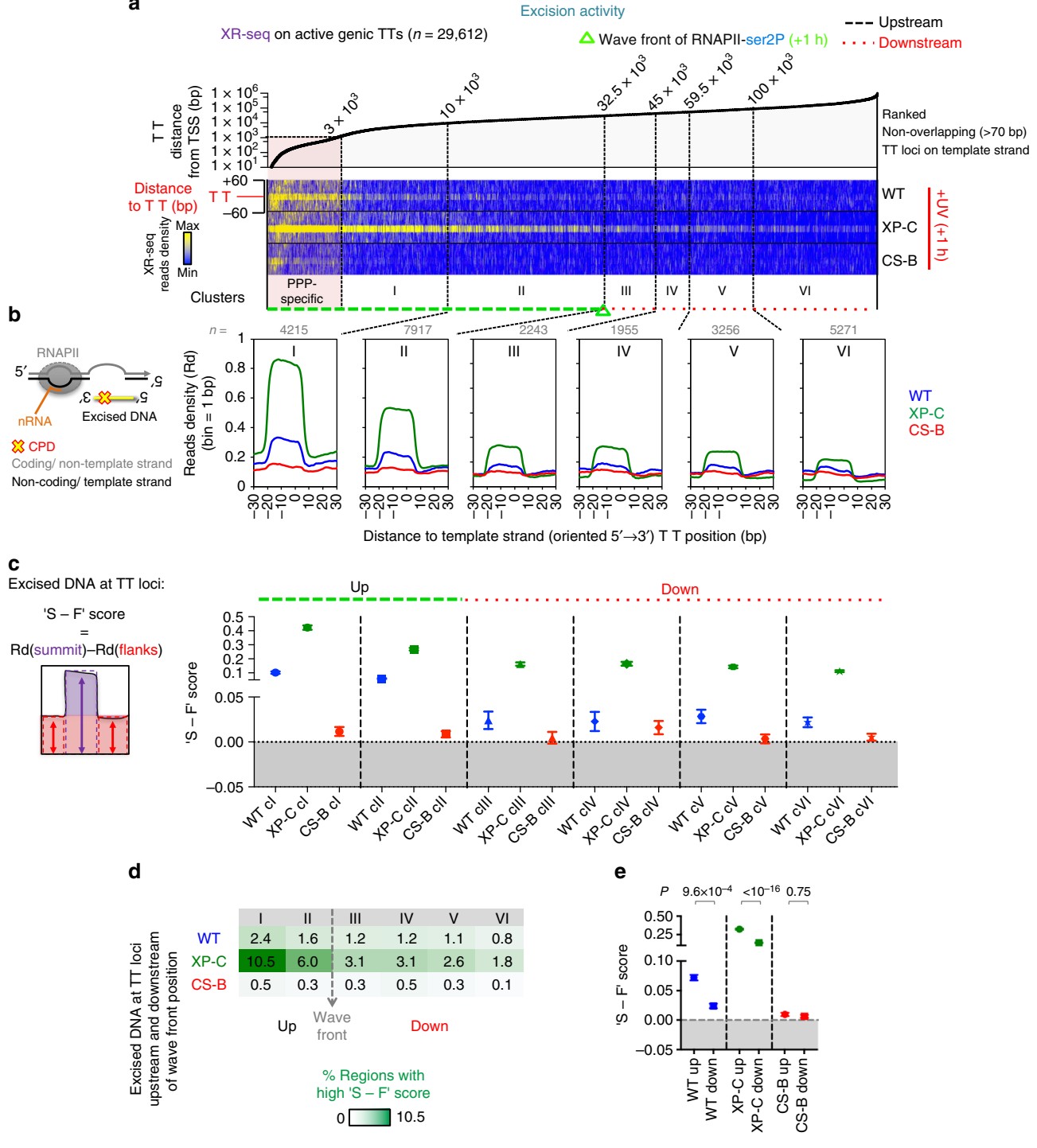

**Fig. 5** NER of DNA lesions is enhanced by the de novo wave-release of RNAPII upon genotoxic stress. **a** Heatmaps illustrating the distribution of excised DNA fragments representative of repair activity at 1 h of recovery in cell lines proficient for TC-NER activity (WT, XP-C), or not (CS-B) (reads from XR-seq study[36]) (see Methods for details). XR-seq reads were aligned around TT loci localised in active genes (Supplementary Fig. 8a) and sorted from left to right by increasing distances relative to TSS. TT loci were clustered (Upstream; clusters I and II, or Downstream; clusters III–VI) relatively to RNAPII-ser2P wave front position at +1 h of recovery (as in Fig. 4a). TT loci near PPP-specific RNAPII signal were not considered for analysis. **b** Average plots of Read densities (Rd) mapped in **a** for individual clusters (n is indicated). **c** Plot showing the average (± SEM) of the difference between Rd at summit and Rd at flanks ('S - F' score) of all regions analysed in **b** (P-value based on two-sided Wilcoxon rank-sum test). **d** Heatmap representing the proportion of regions (per cluster) with high 'S - F' scores (> threshold = average 'S - F' + 3 × SD, as calculated for the control exon start regions, see Methods). **e** Plot showing the comparison of average (± SEM) 'S - F' score for all regions upstream (Up) and downstream (Down) of the respective wave front position, for each cell line (P-value based on two-sided Wilcoxon rank-sum test using BH adjustment)

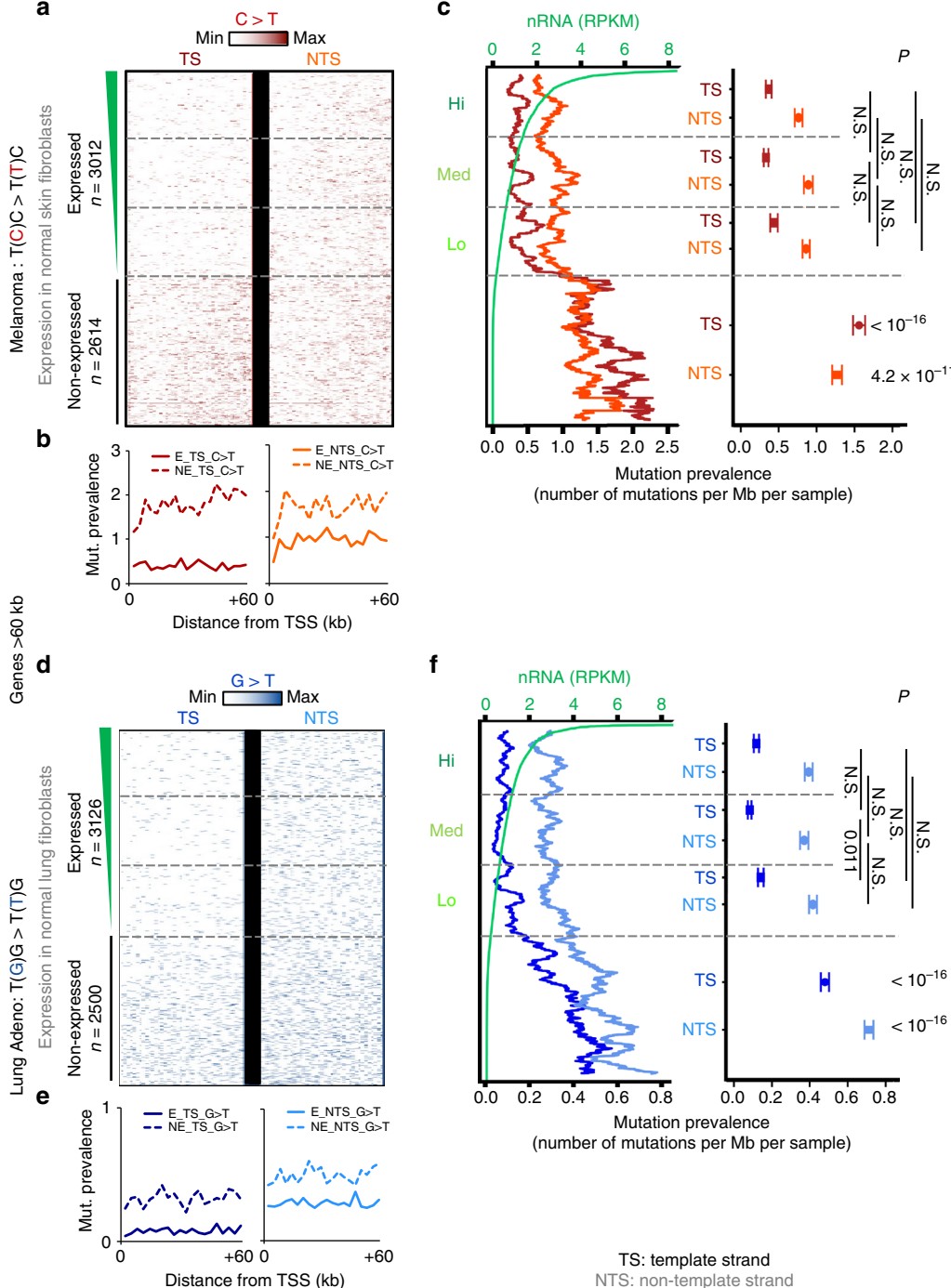

**Fig. 6** Low and uniform mutation rate is detected in all expressed genes in environmentally exposed tumour genomes NER. **a**, **d** Heatmaps showing representative density and location (TSS to +60 kb) of most frequent substitutions associated with **a** UV (C > T) in skin melanoma tumour genomes (extracted from WES data) and **d** cigarette smoking (G > T) in lung adenocarcinoma tumour genomes (extracted from WGS data), for TS and NTS strand separately (see Supplementary Fig. 12a–c and Alexandrov et al.[7]). Genes are stratified by expression levels, as determined by nRNA levels of normal skin and lung fibroblast cell lines, respectively (see Methods). Expressed is labelled as E and non-expressed as NE. **b**, **e** Average mutation prevalence profiles across gene bodies (number of mutations per Mb per sample was also corrected by exon density in the case of WES), for expressed (plain) or not-expressed (dashed) genes. **c**, **f** Left panel: moving average (across expression ranks, $n = 200$) of mutation prevalence (per gene) in **a** and **d**. Corresponding nRNA levels are shown in green. Right panel: comparison of average (± SEM) mutation prevalence (top: pairwise between Hi, Med, Lo expression categories, bottom: between all E and all NE genes, see also Supplementary Fig. 12e, h) for TS and NTS. N.S. indicate non-significant $P$-value (>0.05) (two-sided Wilcoxon rank-sum test using BH adjustment)

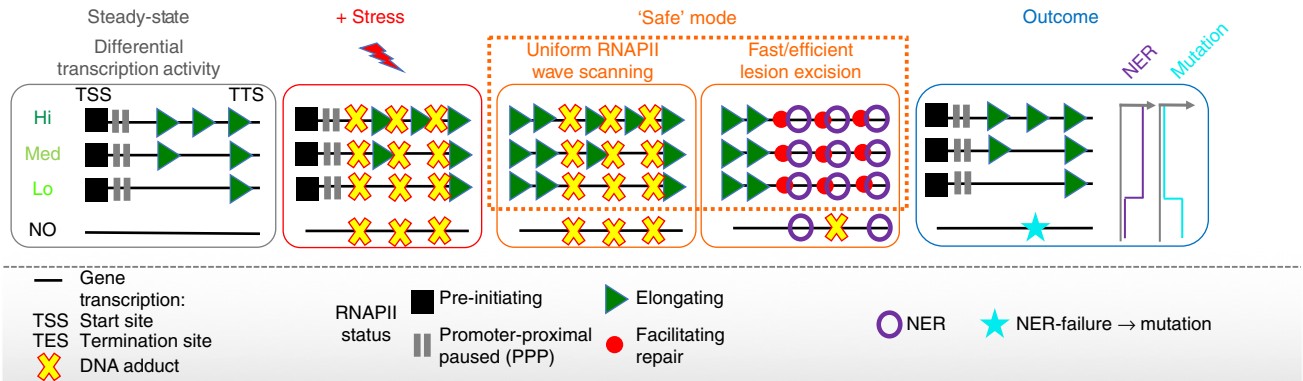

**Fig. 7** Model describing the proposed 'safe' mode of elongation. Upon genotoxic stress, steady-state nascent transcription levels of virtually all expressed genes are adjusted via a transient and uniform de novo wave-release of scanning RNAPII (green triangles) from PPP. This mechanism maximises both speed and probability of the sensing (red dots) and removal (purple rings) of NER-dependent DNA lesions (crosses) throughout the whole transcribed genome. In turn, environmentally exposed genomes are characterised by low and uniform mutation landscape across all active genes in both strands, which sharply contrasts the mutation rates observed in constitutively non-expressed genes. In case NER fails or is not recruited efficiently during stress recovery, unrepaired lesions can provoke error-prone DNA synthesis and result in mutations (turquoise star) during replication

Supplementary Fig. 8c). Hence, we found that co-localisation of RNAPII with potential damaged sites was maximal in clusters located upstream of the wave front position at 2 h post irradiation (Fig. 4a, b: clusters I–III). This result was in sharp contrast with the uniform distribution of DNA photolesions in gene bodies immediately after UV irradiation, as detected by sequencing of the DNA fragments containing CPDs (CPDIP-seq) (Supplementary Fig. 8d) or by a similar recently developed approach[48]. Therefore, our data indicate that the wave-release of RNAPII might participate in establishing the observed oriented pattern for RNAPII stalling at TT lesions.

To quantify the density of RNAPII around TT sites, we calculated the fold change (FC) in read density between UV-treated and NO UV conditions across the regions. We found that at 2 h post UV, up to 68.1 % of the analysed regions were highly enriched with RNAPII (Fig. 4c). To assess more precisely the stalling of RNAPII exactly at damaged TT dinucleotides and to exclude the effect of the wave simply passing-by, we calculated the difference in read density between the summit (S: specific accumulation of RNAPII at the centre of the regions) and the flanks (F: background of the wave passing-by) of TT loci. Comparing these ratios, defined as 'S - F' scores (Fig. 4d), we established that the average stalling of RNAPII at potentially damaged sites is significantly increased upstream of the wave front position (Fig. 4e). In fact, the proportion of lesions recognised by RNAPII (high 'S - F' score) gradually reached a maximal value of 28.2 % of the total number of regions analysed at 2 h post UV in cluster I (Fig. 4d), thus confirming the oriented and beneficial impact of the wave-release on the probability of detecting lesions along gene bodies. In contrast, RNAPII was not specifically stalled around the centre of a control set of intragenic regions such as exon start loci, which are not enriched for dipyrimidines (Supplementary Fig. 9a–c). Accordingly, FC in RNAPII density was lower around exon start loci than around TT loci and RNAPII was not enriched at exon start loci (Supplementary Fig. 9 d, e). These data prove that the signal detected at exon start loci corresponded to the wave passing-by without prolonged stalling. In support of the above findings, the slot-blot analysis of RNAPII ChIP-enriched DNA suggested that RNA polymerases were strongly associated with damaged DNA, especially 2 h post UV (Supplementary Fig. 9f).

We next assessed the importance of the de novo released elongation waves by quantifying the stalling of RNAPII at TT loci after irradiation in DRB-inhibited cells (+DRB + UV). The lack of elongation influx was sufficient to cancel the increase in 'S - F' scores in clusters located upstream of the wavefront positions (Supplementary Fig. 10a), thus significantly eliminating the differences in 'S - F' scores along gene bodies (Supplementary Fig. 10b). Taking into consideration the uniform distribution of pri-elongating RNAPII on gene bodies, prior to UV-irradiation (Fig. 1e), inhibition of the de novo elongation wave-release was sufficient to impair the potential for detecting CPDs located close to TSS. Indeed, pri-elongating molecules located downstream of the damage at the time of stress would not be able to detect those. Consequently, the repair of these lesions would depend exclusively on the slower GG-NER pathway[16]. Together, these data (i) confirm that RNAPII de novo release into productive elongation promotes the accelerated and uniform recognition of DNA adducts across the whole active gene set and across each gene's body and (ii) reveal the molecular causes underlying transcription elongation deceleration by UV lesions in vivo.

**De novo wave-release facilitates DNA repair.** Next, we conducted a meta-analysis of the data reported by the Sancar lab[36] for human fibroblasts with different NER capacities. In particular, we mapped NER-specific excised DNA (XR-seq: fragments produced at +1 h after damage induction) around intragenic TT loci, as described above for the RNAPII reads. In WT and in cells derived from Xeroderma pigmentosum C patients (XP-C), which can efficiently repair DNA lesions in active genes via TC-NER, we discovered that excision activity was enriched for TT loci located close to TSS, upstream of the wave front (Fig. 5a, b, clusters I and II). This was confirmed by the high 'S - F' scores (Fig. 5c, Up) and the increased proportion of regions undergoing repair (Fig. 5d, Up). In accord with the increased efficacy of lesion sensing (Fig. 4) offered by the release of the de novo transcription wave, the repair of TT loci located downstream of the wave front was lower than for the upstream regions, especially in TC-NER-proficient cells (Fig. 5c–e clusters I and II (Up) vs III–VI (Down). In TC-NER-proficient cells, we also found that excision of DNA lesions on the template strand (TS) was generally amplified in active genes compared to inactive genes, with a prominent signal at the 5′ end of transcribed regions (Supplementary Fig. 11a–d). Finally, we demonstrated that in WT cells the repair of active genes occurred equally well, irrespectively of RNAPII density in

gene bodies prior to stress. Although this effect was more striking on the TS, we also detected it to some extent on the NTS (Supplementary Fig. 11a).

Together, repair activity and prolonged stalling of RNAPII at damage sites effectively explain the deceleration of de novo released RNAPII molecules shown above. Interestingly, the excision activity at damaged sites located downstream of the wave spread was moderate, but significant (see Fig. 5, Down). This activity was attributed to DNA damage-sensing and NER recruitment exclusively by the decelerating, yet still bound to chromatin, pri-elongating RNAPII (see Fig. 4, downstream clusters, Fig. 2 and above). In agreement, we found a persisting percentage of regions with high 'S - F' score despite de novo wave inhibition by DRB even in more distal parts of the genes (see Supplementary Fig. 10b).

It is noteworthy that, excision fragments were produced from a large proportion of TT loci as early as 1 h post irradiation (see Fig. 5c, d, clusters I and II) whereas RNAPII molecules recovered from stalling at the same TT loci at 6 h post UV (see Fig. 4d, e, compare +2 h and +6 h). The XR-seq method detects lesions being excised at the time of the assay[36]. In line with this, we found that only a small proportion of regions were being repaired at +1 h, which suggests that completion of the whole repair process may take several hours. Nonetheless, we anticipate that the probability of encountering non-repaired DNA lesions decreases with time. At later time points during recovery, as more lesions were repaired, the balance between lesion-stalled and elongating polymerases was progressively inverted (compare +2 h and +6 h in Supplementary Fig. 9d) and this phenomenon contributed to the eventual reset of steady-state RNA levels and RNAPII distributions at later time points (see Fig. 1d, e, and Supplementary Figs. 2d, 3h, j, k, 4a, c, and 7a, b). We therefore conclude that, as an important part of the process underlying the response to stress, de novo wave-release of elongating RNAPII in active genes promotes both an accelerated and a more uniform removal of transcription-blocking DNA lesions by TC-NER activity on the TS, but also to some extent because of facilitated GG-NER on the NTS (see Supplementary Fig. 11c, d). Finally, although RNA synthesis appears globally decreased, the underlying de novo release of transcribing molecules, which we unmasked, contributes to maximize and ensure prompt recovery from DNA damage via NER.

**Mutation landscapes are shaped in a wave-dependent manner**. One prediction from our findings is that mutation rates calculated for NER-associated signatures should remain low in all active genes independently of their steady-state expression level and of the density of pri-elongating RNAPII. To test this hypothesis, we took advantage of published clinical-relevant genome-wide mutation maps of genotoxin-exposed tissues that have developed cancer. We focused on two data sets[7] previously linked to NER activity, because they display 'T-class' (transcriptional asymmetry, i.e. mutation rate for TS is lower than NTS) -associated base substitutions, which implies that TC-NER limits mutation prevalence[7,10]. Skin melanoma and lung adenocarcinoma tumours are known to show strong probabilities for UV light (C > T) and smoke (G > T)-induced mutagenesis, respectively, especially in non-expressed genes. We selected C > T (or the reverse complement G > A) substitutions, which correspond to the most abundant UV-generated mutations[7,10,11]. Interestingly, although CPDs formation at CC, CT and TC dipyrimidines, is less frequent than at TT dimers[46], they generate more frequently error-prone translesion DNA synthesis and this phenomenon results in C > T mutational events, which are the hallmark of UV-exposed genomes[11,14,49,50]. For smoking-related impairments, G > T (or the reverse

complement C > A) substitutions were selected, because they correspond to the most abundant smoking adduct-generated mutations repaired by TC-NER[7,10,12] (see Methods).

We re-extracted in an expression-, strand- and tumour-specific manner the genomic positions of these NER-related substitutions (Supplementary Fig. 12a–c and Methods). This novel approach enabled the establishment of precise maps of mutational events, revealing detailed insights of the tumours genome mutation landscape. In particular, we were able to determine precisely the localisation of the mutations along gene bodies (Fig. 6a, d and Supplementary Fig. 12d, g). In this respect, the possibility of analysing separately whole-genome sequencing (WGS) and whole-exome sequencing (WES) data for lung adenocarcinoma samples was beneficial. A much lower number of samples sequenced by WGS was required to confidently analyse more mutations (Supplementary Fig. 12b, c). Also, the possibility to map mutations in intron sequences permitted the detection of a more continuous signal density across gene bodies (Supplementary Fig. 12g, WGS). Nonetheless, for melanoma samples, which were only sequenced by WES, we could accurately correct the calculated prevalence of mutations along gene bodies as a function of the underlying density of exon in those genes (Supplementary Fig. 12d and Methods). Thus, in both tumour samples, mutation rates detected along gene bodies were uniform even in more distal parts of gene bodies (Fig. 6b, e). We confirmed that transcriptional strand asymmetry in expressed genes was more prominent than in non-expressed genes[7,10–12] (TS < NTS) (Fig. 6c, f and Supplementary Fig. 12e, h), and for both strands, we detected lower mutation rates in expressed genes (E) than in non-expressed genes (NE) (Fig. 6c, f and Supplementary Fig. 12e, h). More interestingly, we uncovered low and particularly homogenous level of genetic changes in all active genes, in both analysed genomes, independently of their expression level, and for both DNA strands across the whole gene bodies (Fig. 6c, f and Supplementary Fig. 12f, i). Given that the widespread and uniform release of RNAPII facilitates both scanning and opening of DNA and maximises removal of NER-related DNA lesions of all the expressed genes, we conclude that this mechanism impacts significantly on the mutation landscape of various tissues, independently of the exact cause of DNA damage.

## Discussion

In this report, we characterise a novel metabolic function associated with active transcription. We propose that in response to genotoxic stress, threatened cells switch transiently to a 'safe mode' of RNAPII elongation (Fig. 7). This defence mechanism promotes widespread, rapid and synchronous de novo escape of elongating RNAPII from promoter-proximal regions into the gene bodies of virtually all active genes, which represent almost 40% of the genome[51]. By maximising the entry of RNAPII in all active genes, cells benefit from an accelerated and uniform DNA damage-sensing by RNAPII on the TS, and to a lower extent on the NTS, regardless of damage location, gene length and steady-state levels of transcription. This functional advantage is dependent on the release of elongation waves, as it is lost upon transcription inhibition (irradiation in the presence of DRB). In line with this, a boost in NER activity is observed at lesions located within the range of the newly released RNA polymerases. Our data also reveal clinically relevant consequences of this unanticipated defence mechanism. The somatic mutation landscapes of cancer genomes, such as melanoma and lung adenocarcinoma exposed to UV irradiation and cigarette smoke respectively, consistently display low and homogenous mutation rates in all expressed genes. Finally, our results substantiate the importance of the promoter-proximal

transcription regulatory step and explain why cells can benefit from priming paused RNAPII in such an extended proportion of active genes.

Our approach provides clear quantitative views on the distribution patterns and dynamic changes in the abundance of the three main isoforms of RNAPII (from pre-initiation complex formation to entry into productive elongation) and nascent RNAs in response to low doses of UV irradiation.

Mechanistically, the unanticipated widespread productive elongation boost, fed by the entry of new RNAPII molecules into gene bodies, is compatible with previous findings showing that, while in normal conditions P-TEFb function is restricted by the sequestering effect imposed by 7SK snRNP inactivating complexes[52], UV irradiation promotes a rapid increase in the pool of active P-TEFb molecules in the nucleus[35]. We elucidate the functional consequences of this activation. Upon stress, the unleashed P-TEFb kinase activity (via cdk9) leads to extensive hyper-phosphorylation of RNAPII CTD (into RNAPII-ser2P)[29,53] and accounts for the global and synchronous transition into productive elongation observed in virtually all active genes (Fig. 1c–e). Removal of DRB further expands the wave-release effects observed in physiological conditions (without DRB, Supplementary Fig. 6b–d and see Fig. 1d, e), which demonstrates that the amplitude of the wave-release is dependent on the proportion of PPP sites readily occupied by paused RNAPII. Our model thus reveals important insights in the specific nature of a predicted, yet not characterised, central role of P-TEFb in the DNA damage response that was suggested based on recently published proteomics analyses[54].

The identification of a UV-induced widespread release of elongating RNAPII also clarifies the findings of previous reports, which detected in response to UV irradiation, but interpreted differently: (i) increased levels of RNAPII in most active genes (see Fig. 4 of Gyenis et al.[38]), and (ii) increased nascent RNA levels at the beginning of genes[44,45]. Challenged cells enable robust RNAPII-dependent accelerated 'lesion scanning' activity in all genes essential to their homeostasis. De novo released RNAPII molecules guarantee that lesions located very close to TSS or in less frequently expressed genes will also get promptly repaired. By transiently adjusting gene expression at the level of PPP-release, cells activate a pre-wired program of 'safe' mode elongation and limit potential biases associated with inherent stochasticity of the transcription initiation step[24,55]. In addition, sending waves of trailing RNAPII molecules throughout the transcribed genome could allow for the next lesions in the gene body to get efficiently detected and repaired even in the case of dissociation from chromatin of the initial damage-detecting and NER-triggering RNAPII molecules[39,44,56]. In turn, this pathway will ensure a smooth transcription restoration process in expressed genes after repair has been completed regardless of gene length and expression level.

Functional genomic cross-comparisons of our RNAPII ChIP-seq data with NER-specific excised DNA data (XR-seq[36]) demonstrate a striking parallel between the global wave-release of damage-sensing RNAPII molecules and the increased repair efficacy upon genotoxic stress in all active genes. As shown in Fig. 5, the proximal vs distal difference in excision-reads observed in both WT and XP-C cells 1 h post UV is due to de novo wave propagation of RNAPII and to the substantial increase in the probability of transcription-dependent repair. As previously reported[36], we note that XP-C cells show an increased XR-seq rate of UV lesions in comparison to WT cells. This might be partially due to the lack of repair in intergenic regions and non-template strands in XP-C cells. Thus, coverage of XR-seq reads is limited to a smaller part of the genome and a substantial over-estimation of the enrichment in read density in genic region is

expected. Additionally, in these cells, the absence of GG-NER pathway might affect the probability of the available core NER factors to be recruited at transcription-blocking lesions.

While our proposed mechanism facilitates TC-NER by fast sensing of transcription-blocking lesions by RNAPII on the TS of active genes, we note that the very progression of the transcription wave may result in a more open chromatin environment and could indirectly augment the repair rate of the NTS by GG-NER. Indeed, the role of chromatin structure in governing the repair efficiency is indicated by earlier studies showing that repair in the NTS of active genes is faster than in inactive genes[57]. Corroboratively, in WT cells, lower levels of repair could be detected in the NTS of inactive genes than of active genes (Supplementary Fig. 11c). In line with this, a recent study, showed that open chromatin state facilitates repair by GG-NER along most of the length of DNAse hypersensitive (DHS) regions[58–60]. We thus conclude that, increased opening of chromatin can enhance both TC-NER and GG-NER accessibility to damaged DNA.

Maps of somatic mutations of genotoxins-exposed cancer genomes such as melanoma and lung adenocarcinoma[7] have previously been demonstrated to contain NER-specific signatures (see refs. [6,14]). Indeed, these tumours originate from skin and lung tissues that may have been exposed to environmental stress such as UV-irradiation and tobacco smoke, which trigger NER activity. By extracting the substitutions in a tissue-, strand- and expression-specific manner we found that, the probability of mutagenesis remains low throughout the TS of active genes regardless of these genes' level of expression. To some extent a reduced mutation rate is also observed in the NTS of active genes, thus confirming better efficacy of both NER subpathways.

In nature, differential exposures to UV light may also provoke overcoming a signalling threshold at which the response we present here can be activated. It can be speculated that cells can determine when dangerous levels of exposure have been reached to justify a rapid response. In addition, the levels of exposure requiring the activation of such 'safe' mode might be lower for cells that replicate more often, as these will need to benefit from accelerated repair before they can divide safely, thus preventing potential cancer stem-cell formation (see Liakos et al.[8] for review).

Our results demonstrate that the global release of elongating RNAPII waves promotes NER efficacy and can in turn preserve genetic accuracy. We thus anticipate that general deficiencies in transcription elongation and/or TC-NER will limit such genome-protective effects. In line with this, during the past few years the biomedical field has concentrated on the investigation of novel markers ruling DNA damage response pathways. This significant effort fuels the discovery of new potentially 'druggable' targets and could facilitate patients' access to safe and precise cancer therapies[5,61]. In fact, several studies suggest that drug- or genetically impaired NER function favours cisplatin response, one of the most potent anticancer treatments[5], and that NER factors are often upregulated in solid tumours with resistance to chemotherapeutics (see review[8]). In that sense, we believe that inhibiting the ability of tumour cells to boost transitioning of RNAPII into productive transcription elongation, while promoting genotoxic stress, may also improve cancer therapeutics.

## Methods

**Cell culture and cell treatments**. Cell line used in this study was VH10 normal human foreskin fibroblasts (VH10) hTERT immortalised as previously described[62]. Cells negative for mycoplasma (Jena Bioscience) were propagated under standard culture conditions in Dulbecco's Modified Eagle's Medium (DMEM, Thermo Scientific), supplemented with 10% v/v fetal bovine serum (FBS, Thermo Scientific), 1% v/v penicillin–streptomycin (Thermo Scientific) and in G418 (25 μg ml$^{-1}$ final, AppliChem) for selection, in a humidified incubator at 37 °C and 5% $CO_2$. When applicable, after one week in culture confluent cells were synchronised by

72 h of serum-deprivation (starving in DMEM + 0.5% FBS) followed by release for indicated times in complete medium (released in 10% FBS) before treatment to rule out cell cycling heterogeneities, to increase the proportion of cells in G1 at time of UV irradiation, and to boost steady-state levels of transcription. Cells were UV-C irradiated (at 254 nm, TUV Lamp, Philips) at indicated doses. Transcription Inhibitor 5,6-dichloro-1-β-D-ribofuranosyl-benzimidazole (DRB, Calbiochem) or other chemicals were added directly in the growth media, unless otherwise indicated, at concentrations and times specified in the figures.

**ChIP-seq.** Crosslinked ChIP was carried out from at least two independent cultures of cells per condition, as previously described[55] with minor modifications. Antibodies used are as follows: Pan-RNAPII (Rpb1 N-terminus) (EPR1509Y; ab76123, lot# YI03010C), RNAPII CTD (phospho-S2) (ab5095, lot GR231750-2) from Abcam; RNAPII CTD (phospho-S5) (3E8; 04-1572-I, lot: 2395817) and RNAPII CTD (hypophosphorylated) (8WG16; 05-952, lot: 2262610) from Millipore (see Supplementary Methods). Final ChIP DNA was quantified on a Qubit 2.0 Fluorometer (dsDNA HS Assay Kit, Thermo Scientific) and ChIP specificity was checked by qPCR analyses performed with 10–100 pg of ChIP and Input DNA in duplicate reactions with qPCRBIO SyGreen mix (PCR Biosystems) on Roche Light Cycler 96 instrument. We validated at least two independent ChIP replicates by ChIP–qPCRs. If individual ChIPs showed enrichment in expected genomic regions (as determined by qPCR ΔΔCt method between positive and negative loci normalised with inputs, primers sequences are available upon request) DNA were combined to reach ~1–10 ng of material. ChIP and Input DNA were then subjected to library preparation for high-throughput sequencing (see below).

**Nascent RNA sequencing.** Nascent RNA (nRNA) was isolated using the Click-iT Nascent RNA Capture Kit (Life Technologies), according to the manufacturer's protocol. In brief, two independent cultures of cells were treated, as indicated in the legends, with 5-ethynyl uridine (EU, 0.1 mM) for 10 min, at specified time points, before total RNA was isolated with Trizol as described in the previous section. 5 µg (quantified by Nanodrop) of DNAse-treated (Turbo DNAse, Ambion) total RNA was biotinylated with 0.5 mM biotin azide and bound to streptavidin beads. Nascent RNA (nRNA: EU-containing RNA captured on the beads) was fragmented and used for first strand cDNA synthesis before being subjected to second strand cDNA synthesis as described in the previous section. Eluted DNA was size-selected and purified with Ampure XP beads before library preparation.

**DRB-release (preDRB-) and DRB-inhibition (DRB-) experiments.** For preDRB-nRNA-seq, two independent cultures of cells were pre-incubated with DRB (100 µM) for 3 h to drain elongating polymerases from gene bodies based on previously described protocols[22,63–65]. Nascent RNA was labelled with EU for at least 10 min before RNA extraction at indicated times of recovery with Trizol reagent as described above. For $t = 0$ total RNA was isolated immediately after EU labelling. For the other time points, at time 0, DRB was washed-out by PBS and mock (NO UV) or UV-irradiated (20 J m$^{-2}$). PBS was replaced with fresh complete medium, to allow for normal recovery, before RNA was isolated at the indicated time. For DRB inhibition (DRB-ChIP-seq), two independent cultures of cells were incubated with DRB (100 µM) 10 min before UV irradiation (20 J m$^{-2}$). After irradiation, cells were left to recover for indicated times in the presence of DRB before crosslinking, chromatin isolation and ChIPing for RNAPII-ser2P (as above).

**Illumina sequencing and library generation.** Library preparation was performed essentially according to standard Illumina protocols and as previously described[55] with minor modifications. DNA-modifying enzymes were purchased from NEB. dsDNA was end repaired, 3′-end adenylated and ligated to barcoded (for multiplexing) Truseq (synthesised by IDT) or NEBNext adapters according to manufacturer's guidelines. Samples were PCR amplified by insuring reduced amplification bias (optimal cycle number was determined by qPCR using the Illumina P5 and P7 primers) and size-selected (200–500 bp) through double selection by Ampure XP Beads as previously described[66]. Libraries were assayed on a BioAnalyzer (High Sensitivity DNA kit, Agilent) and next-generation sequencing was performed at Genecore-EMBL, using the Illumina HiSeq 2000 for single end 50 bp reads.

**Reads alignment and normalisation.** For all next-generation sequencing analyses, we developed in-house scripts and pipelines to automate and analyse the data consistently (see below for details). Codes are available upon request. Sequenced data and generated wig profiles are available on Gene Expression Omnibus (GEO) (Accession ID: GSE83763). Briefly, reads were quality trimmed, Illumina adaptors were removed and filtered reads were aligned uniquely to the GRCh37/hg19 reference genome with specific parameters for each experiment (see Supplementary Methods for details). Sequenced, mapped and normalised read stats are depicted in Supplementary Data 1.

**Gene transcripts and exons annotation.** The 54,669 isoforms of release 72 of RefSeq collection were filtered as follows: (i) neighbouring TSSs within distance of less than 500 bp were collapsed and the longest isoform was kept for each gene as

described previously[64]; (ii) neighbouring isoforms with inter-TSS distance <1 kb were excluded from the analysis to avoid potential artefacts of analysing reads from nearby gene promoters and of merging distinct genes with overlapping promoter-proximal regions; (iii) only protein coding and lncRNA genes were kept for analysis according to the NCBI RefSeq biotype specification. Consequently, the analysis resulted in n = 19,775 curated genes.

For constructing our custom exon annotation, we merged (bedtools merge -d 1) all the overlapping genes for each strand separately. Genes with overlapping exons encoded in opposed strands were not considered in order to avoid multiple strand references. Consequently, the analysis resulted in n = 164,896 merged exons out of n = 205,798 individual exons.

**Gene expression status definition.** Constitutive gene expression status in the steady state was determined before UV irradiation as follows: Briefly, using MACS2[67] we called ChIP-seq peaks for RNAPII-hypo, -ser5p and -ser2P ChIP-seq in steady-state (NO UV) conditions, using only non-redundant reads[68] comparing to Input and '-no-model' option. The list of significant ChIP-seq peaks (fdr < 0.05 and fold change > 1) is presented in Supplementary Data 2). Peaks of RNAPII-hypo, -ser5P and -ser2P overlapping with gene promoters (−250 bp to +100 bp around TSS) were analysed to establish gene activity status. Genes containing RNAPII-ser2P peaks and were required to show read density (Rd) at promoters >0.7 rpm in accordance with previous reports[69] and were considered active (n = 8954) (density in gene promoter-proximal regions ranging from −250 bp to +100 bp around TSS was abbreviated as 'Dp' and was expressed in reads per million (rpm)). Poised genes were defined as the union of RNAPII-ser5P and −hypo genes that did not overlap with -ser2P peaks (n = 953). The rest of the genes were considered as inactive (n = 9868). All genes annotation and expression status information is presented in Supplementary Data 3. RNA-seq activity status was determined as previously described using as threshold the lower value between peaks in the log$_2$ RPKM kernel density distribution plot[70].

**Promoter escape indexes and ΔEI.** Promoter escape indexes (EI) were calculated for curated genes, as previously defined with minor adjustments[69,71], by taking the average coverage in rpm in the gene body (density in gene body was abbreviated as Db and ranged from 101 bp to 2 kb downstream of TSS or 101 bp downstream of TSS to TTS for genes larger or smaller than 2 kb, respectively) divided by the average coverage in the promoter-proximal region (Dp), as defined in the previous section. EI changes (ΔEI) between conditions were calculated by dividing EIs as indicated in legends, and were used to assess differential RNAPII Promoter Escape upon treatments. The proportion of genes with increased escape after treatment are shown in percentages and Chi-square test ($\chi^2$) determines if observed number of genes with ΔEI > 1 differs from expected value purely by chance. PCC were calculated in Excel or R statistical package accordingly.

**Read densities heatmaps and average plots.** ChIP-seq and nRNA-seq data were subjected to read density analysis after normalisation of all samples per experiment (see above). For heatmaps and average density profiling and differential enrichment profiling, SeqMINER 1.3.3[72] or DeepTools (http://deeptools.ie-freiburg.mpg.de/) were used to extract read densities at genomic regions (metagenes or TT loci) of interest as indicated in the figures. Heatmaps were generated directly in the software from matrixes of binned read densities (bin size is indicated in the figures) for all considered individual (n) items (metagenes). Matrices were also exported to excel for (i) plotting average density profiles (smoothing achieved by a moving window of the bin size as indicated), (ii) for determination of wave front and backend positions and (iii) for calculation of elongation rates (see following section). When applicable the matrix of read densities was sub-divided in categories (deciles or clusters, as indicated in the figure legends) and the average density profiles of each cluster were considered. For determination of fold changes profiles, the number of reads per bin for a given sample were divided by the number of reads in the indicated control sample and expressed as Log$_2$ FC.

**Wave position and elongation rate estimation.** The front (or leading edge) and the backend (lagging edge) position of the RNAPII or nRNA wave at each time point was computed as the position at which the average densities of n considered metagenes crossed an arbitrary threshold representing the transition point indicated in the figures. These positions were calculated relatively to the TSS and expressed in kb. Average elongation rates (kb min$^{-1}$) were calculated from the composite average plots as previously described[64,65]. Importantly, this method accounts for variability across metagenes (n) given the large number of measurements and the distance covered between two time points.

**Genome-wide identification of TT and exon start loci.** TT dinucleotide loci corresponding to the motif XTTX, where X = {A, C, G}, were called within active gene borders (TSS to TSS + 10$^6$ bp) on the non-coding strand and further filtered for TT loci with the distance to next/previous item (non-overlapping) greater than 70 bp. These corresponded to n = 29,612 active genic TTs (Supplementary Data 3). RNAPII-ser2P reads (see normalisation section above) and excised fragments reads (see next section) were used to generate density plots centred at TT loci (−400 bp to +400 bp from TSS and −60 bp to +60 bp from TSS, respectively) as described in

'Read densities heatmaps and average plots' section for all cluster as defined in figure legends.

Exon start loci positions were extracted from our custom exon annotation, we filtered a representative subset of regions by selecting exons id number equal to $2x$, where $x$ could be any number (grep 'e2').

**Nucleotide excision repair data meta-analysis.** The strand-specific genome-wide maps of nucleotide excision repair of the UV-induced DNA damage (CPDs), available for three different cell lines (i) wild-type (WT) NHF1 skin fibroblasts, (ii) XP-C mutants, lacking the global genome nucleotide excision repair mechanism (GG-NER-deficient, TC-NER-proficient) and (iii) CS-B mutants lacking transcription-coupled repair (TC-NER-deficient), were obtained from Hu et al.[36]. Raw data files were downloaded from Gene Expression Omnibus (GEO, accession number: GSE67941). Meta-analysis involved that read counts were normalised. We computed heatmaps matrices and average reads density plots as described in the section 'Read densities heatmaps and average plots'. Around TT loci, we analysed both strands together, as defined in figure legends, and as described in the previous section. We also computed average reads density plots, for both strands separately for analysis over gene bodies as explained for ChIP-seq above.

**Quantification of reads around specific loci.** To estimate the general enrichment of reads in a region between conditions, we relied on $Log_2$ FC in Rd between irradiated (+UV) and steady state (NO UV) were calculated within the observed regions (TT loci and exon start loci, see above) for each cluster. For multiple comparisons, normal distribution was assumed as supported by the $log_2$ FC values. One-way analysis of variance (one-way ANOVA) was performed to test for significant differences between examined conditions, while pairwise $t$-tests with Benjamini–Hochberg (BH) adjusted $P$-values were also performed as post hoc tests, to identify the pairs that differed.

To estimate more specifically the stalling of RNAPII (RNAPII-ser2P ChIP-seq reads) or the excision activity (XR-seq reads) we calculated the difference ('S - F' score) in Rd between the summit (S: specific accumulation at the centre of the regions) and the flanks (F: background expected from the wave passing-by) for TT loci and for exon start loci.

For multiple comparisons, normality was not assumed, and the non-parametric Kluskal–Wallis one-way analysis of variance was performed to test for significant differences between examined data sets. Pairwise Wilcoxon rank-sum tests with BH adjusted $P$-values were also performed as post hoc tests, to identify the pairs that differed.

The percentage of regions in a cluster displaying FC or 'S - F' scores larger than an arbitrary threshold were calculated in TT loci and in exon start loci references. The FC threshold was set to be FC = 2, while the threshold for 'S - F' score was estimated from the exon start data and set to 'S - F' = average[S - F]$_{exon\ start}$ + $3 \times$ SD [S - F]$_{exon\ start}$. Chi-square ($\chi^2$) test were calculated for pairwise comparisons (TT vs exon start) of numbers of FC and 'S - F' score above threshold in matching clusters. $P$-values reflect if observed values differ from expected value purely by chance.

**Mutation rates analysis.** The genome-wide maps of validated mutations characterised in available human melanoma and lung adenocarcinoma tumours[7] were downloaded from ftp://ftp.sanger.ac.uk/pub/ cancer/Alexan-drovEtAl. We considered all the mutations listed in the files name_-clean_somatic_mutations_for_signature_analysis.txt where name = 'Melanoma' or 'Lung_Adeno'. We analysed separately WGS and WES data separately when possible. We filtered the substitutions according to the scheme presented in Supplementary Fig. 12a–c. Briefly, C > T (or the reverse complement G > A) substitutions were selected because they correspond to the most abundant UV-generated mutations[7,10,11]. Noteworthy, CC, CT and TC CPDs, although less frequent than TT dimers[46], generate more frequently error-prone translesion DNA synthesis, which results in more C > T mutational events[11,14,49,50]. For smoking-related impairments, G > T (or the reverse complement C > A) substitutions were selected because they correspond to the most abundant smoking adducts-generated mutations[7,10,12]. In order to extract the trinucleotide context of these selected mutations, we retrieved from hg19 fasta file the flanking bases around each selected mutation. We then filtered all the hits corresponding to the most frequently characterised trinucleotide events for UV (T(C)C > T(T)C (and G (G)A > G(A)A)) and smoking (T(G)G > T(T)G (and C(C)A > C(A)A)) in Melanoma and Lung Adeno data sets, respectively and generated BED files. As mutational events derive from unrepaired cytidine adducts (UV[50]) or unrepaired benzo [a]pyrene Guanine adducts (smoking), we were able to determine whether the mutations were on the template strand (TS) or on the non-template strand (NTS) (Supplementary Fig. 12a). To achieve this, we intersected the mutation bed files generated above with our transcript annotation (see Supplementary Methods), which was preliminarily separated in two lists regrouping the '+' or the '−' genes.

Gene expression levels RPKM were extracted from nascent RNA-seq data generated for normal (non-stressed) skin and lung human fibroblasts (BRU-seq from HF1 cells[44] and GRO-seq from MRC5VA cells[45], respectively). We filtered out the genes that were not present in our custom annotation and we ranked genes by decreasing expression values. Quantiles of expression were determined arbitrarily. Non-expressed (NE) genes corresponded to the genes with RPKM

values below the threshold established in Kernel distributions as outlined above (see 'Gene transcripts annotation and gene expression status definition' section). The remaining expressed (E) genes were separated in three categories of the same size where Hi, Med and Lo denote high, medium and low expression levels, respectively.

We mapped the selected mutations, as we did for sequencing reads above (see 'Read densities heatmaps and average plots' section). Mutation frequencies were calculated by averaging the number of mutations detected in all analysed samples, for WES or WGS data sets, over the considered gene regions and by normalising the value to what it would be in a region of 1 Mb of DNA (mutation prevalence = number of mutations counted per Mb and per sample). Given the non-linearity of WES data, which is due to the non-homogenous density of exons along gene bodies, we corrected mutation prevalence scores in each considered window as a function of the relative exon density measured in the considered regions (see Exons map in Supplementary Fig. 12d, g, which shows how our custom exon annotation maps over gene bodies in Seqminer analysis). Mutation heatmaps, density profiles and frequencies were extracted, as above for sequencing reads, in a strand-specific manner (see 'Read densities heatmaps and average plots' section). For smoothing prevalence plots of Fig. 6c and f, we used a moving average method of a width of 200 genes. To improve pairwise comparisons of prevalence scores, and to deal with the inherent sparsity of the data (mutations are rare events), we cumulated the mutations found in groups of 15 genes within each expression cluster. As the distribution of the data was non-normal, we performed pairwise comparisons of mutation prevalence by two-sided Wilcoxon rank-sum test using BH adjustment. Non-significant (N.S.) difference among means was called if $P > 0.01$.

**Data availability.** The data reported in this manuscript have been deposited with the Gene Expression Omnibus under accession code GSE83763. Data are available from authors upon reasonable request.

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

## Acknowledgements

We thank Pantelis Hatzis, Panagiota Karagianni, Alan R. Lehmann, Eva Mantouvalou, Iannis Talianidis, Dimitris Thanos, Mihalis Verykokakis and members of the Fousteri lab for critical discussions and reading of the manuscript, and Panagiotis Moulos for help and access to under-development scripts. We also are grateful to Vladimir Benes and the Genecore facility (EMBL, Germany) for the sequencing support. This work was funded by a European Research Council grant to M.F., Agreement-309612 (TransArrest).

## Author contributions

M.F. and M.D.L. designed the study and wrote the paper. M.F. directed the study, obtained financial support and was responsible for interpretation of the results. M.D.L. performed the experiments with the help of K.Z.N.-Z. (mRNA-seq and nRNA-seq) and A.L. (western-blots, slot-blots and flow cytometry). M.D.L. and D.K. performed the statistical and bioinformatics analyses. D.K. and K.Z.N.-Z. contributed equally to the

paper. All authors discussed the results, reviewed, commented and approved the final version of the manuscript.

## Additional information

**Competing interests:** The authors declare no competing financial interests.

