## [Peer Review File · Nature Communications]

Reviewers' Comments:

Reviewer #1:

Remarks to the Author:

In this manuscript, Lavigne et al, examine the relationship between transcription and DNA repair after genotoxic stress (UV damage). The results are intriguing in that they see a shift in polymerase distribution after UV stress as polymerase at promoters become released into productive elongation. The shift is believed to be due to the release of P-TEFb from its repressive complex into a form that can activate paused polymerases. Although not shown here, UV-induced shift of P-TEFb has been shown previously. Using DRB to synchronize transcription elongation, they go on to show that the rate of transcription is slower after UV induction, and that the wave front of polymerase correlates well with location of TT-lesions (measured by CPDIP-seq in this study) and repair sites (measured by XR-seq in another study).

This suggests that the increased polymerase in the gene are responsible for recognizing lesions and stimulating repair mechanisms. Consistent with this, they go on to show that active genes in both lung and skin cancer samples have lower mutation rates in expressed genes. They suggest that the release of PPP results in protection for all active genes regardless of their original transcription levels. They are quite a few datasets here and the production and analysis looks to be of sufficient quality, for the most part. The conclusions seem to be well-supported by the data and they also make sense biologically. I recommend that the authors address the following issues before this work is considered for publication.

Major points:

1) The Svejstrup lab recently published GRO-seq data after DRB synchronization and UV treatment: <http://dx.doi.org/10.1016/j.cell.2017.01.019> . They also show that transcription rate is reduced. The authors should be citing this article as well as addressing how their data compare.

2) They show that the wave front of polymerase correlates well with positions of NER recognition and repair. To show a more causal role for transcription here, could the authors show the rate of recognition and/or repair if UV-stress is reduced in the absence of transcription elongation (i.e. in the presence of DRB?) They may already have this data.

3) How well do their transcription rate estimations match between ChIP-seq and nRNA-seq?

4) There is a clear UV-dependent increase in transcription in the genomic data, however, in cells (Figure S4b), there is a reduction at 2hr. Please explain.

5) Page 7: "Interestingly however, when cells were exposed to UV, nascent transcription recovery could still be detected to a significant degree, suggesting that UV damage does not prevent de novo RNA synthesis."

Comment: As mentioned point 1, this observation has been reported, and should be addressed here.

Minor points:

1) Figure 5 and 6 seem redundant and could possibly be condensed and joined if space is of concern.

2) For clarity on the advantage the described mechanisms confer on active genes, they could

incorporate what happens to no-active genes into model in figure 7.

3) Bottom of page 9. Use of 'pre-elongating' is confusing. Consider rewording.

4) Page 10: "A prediction from our findings would be that, the mutation rate of NER associated signatures should remain low in all active genes independently of their expression level."

Comment: No comma necessary

Reviewer #2:

Remarks to the Author:

The authors unveil a global release of paused polymerase in response to UV treatment. This phenomenon is extremely interesting and as the authors suggest, could assist in detection of DNA damages and reduce mutagenesis.

The described findings are novel and of high importance to people coming from a broad range of fields – including of course researchers in the repair and transcription communities. Following the different isoforms of PolIII after UV is an elegant and complicated experiment – and I find the comprehensive characterization of PolIII occupancy together with nascent RNA synthesis convincing and of great importance.

The majority of my comments relate not so much to the quality of the data but to it's interpretation.

Major comments.

1) To what degree are the experiments quantitative?

The finding that RNA PolIII release from promoters in an elongation wave is very convincing. In general, however, I find the more "quantitative" claims the authors make to be less convincing.

Comparing different ChIP-seq data sets in a quantitative manner is problematic. The authors rely on Chromatin fraction- and ChIP-western blots to show the overall changes in the abundance of different isoforms – but to make these western blots truly convincing I would like to at least see quantification (and in replicates).

The escape indices of the polymerase is defined as a ratio between polymerase at the gene body and the promoter. The changes in escape indexes can be a combination of switching from paused to elongating, but can also be a cause of differential dissociation from DNA. The authors claim that EI for low expressed (highly paused) genes increase more than EI for highly expressed. If this is the case, and there is indeed an "override" of the transcriptional program – You would expect the RNA PolIII levels after UV to NOT correlate with the previous steady state transcription levels. Is this the case?

And a more general note – this entire mechanism is still limited only to those genes that were expressed to begin with – and so not a complete override.

2) Do polymerases fall off or continue after encountering the damage?

A longstanding question in transcription-coupled repair has been whether RNA polymerases that encounter damage dissociate or continue after repair. The recent paper by Andrade-Lima et al had brought forth evidence for dissociation. Here, the authors performed a very sophisticated experiment in which they used DRB to eliminate actively elongating RNA polymerases from genes and follow transcription of the de-novo released polymerases by RNA PolIII-Ser2 ChIP. These experiments indicate that after release, these RNA polymerases continue throughout the genes

and do not dissociate.

The 2015 paper followed nascent RNA sequencing and not RNA polymerase occupancy after UV. Since Andrade-Lima et al saw lower levels of coverage at 3' ends of genes, their conclusions were that RNA polymerases fall off as they encounter damages. After DRB conditions, Lavigne et al also follow nascent RNA transcription – and claim that since transcription does recover it indicates UV damages do not prevent de novo synthesis.

This part of the interpretation is less clear to me. It does appear that transcription is lower at the 3' regions of the genes – even at 2h where at 1h transcription was evenly distributed without damage (Figure 2e,f). The authors explain this by the slower progression of polymerases over a damaged template. The prediction would be that at later time points, as the polymerase progress, RNA synthesis at the 3' end is recovered – but such a time point is not shown.

3) How does the post-UV transcription relate to the underlying damages and their repair?

The authors performed experiments to map CPDs following UV in a genome-wide manner using CPDIP-seq. These experiments are not thoroughly validated and truthfully it is unclear if they were successful. However, previous works to map CPDs genome wide (authors should also quote Teng Y et al 2011, doi: 10.1093/nar/gkq1036) have shown that generally, damage profiles are dictated by the TT frequency- and so the approach the authors took of simply following patterns centered on TT nucleotides is valid and acceptable.

The authors compared the pattern of de novo elongating RNA polymerases to both TT density and the patterns of repair measured by XR-seq. However, wouldn't a similar correlation be expected to total RNA polymerase density, or the steady state levels of RNA polymerase prior to UV? The quantitative nature of the result is not clear. What proportion of the damages are recognized by these de novo released polymerases and how many were recognized by the polymerases that were actively elongating at the time of UV? Perhaps an experiment comparing repair after DRB vs in non-DRB treated cells could delineate the contribution of these newly released polymerases?

In the interpretation of XR-seq data, the authors should bear in mind that this data too is not quantitative. In wild type cells, while very strong XR-seq signal is observed at 1h – The majority of the lesions are still there– and for hours after. It is problematic to assess the quantitative differences in overall repair levels between WT/XPC/CSB cells from the XR-seq patterns. Furthermore, XR-seq does not measure total repair level, but only the repair level at the 10-30min leading to that time point. Therefore it is not at all clear that the wave of RNA polymerization is clearing up damages or contributing significantly to the recovery from damage.

The authors show that transcription coupled repair occurs equally well on all expressed genes despite differences in the escape indexes (Fig 4a). However, it was previously shown that transcription coupled repair correlates to the RNA levels from the genes prior to UV treatment (Hu et al, 2015). This highlights the importance in incorporating the initial transcription levels in the analysis.

4) How does the phenomena of de novo release of elongating RNA polymerase relate to mutagenesis?

In their analysis, the authors calculated the frequencies of cancer-linked mutations on the transcribed and non-transcribed strands of genes. I have two major reservations about their analysis.

First: The mutations analyzed were C>T - while the damages that were discussed all along were TT dimers.

Second: The Alexandrov et al. mutation data set was produced primarily by exome sequencing. Therefore, in the analysis only exons should be taken into account and not the entire transcript.

Lastly, and likely beyond the scope of this paper: Do the authors have any indication if this phenomena is specific to UV or occurs in response to other DNA damages or even other stress types?

Minor comments:

1) Showing that DNA from RNA PolII ChIP is enriched in DNA damage 2h after UV: The dot blot shown in Fig 8e does not contain a negative control blot, and I understand blotting 200pg of damaged DNA is difficult. I would appreciate including the control (undamaged) DNA in the blot, as well as adding error bars to the quantification.

2) Please make figure labels a bit larger. Figure 2C was impossible to read in print.

3) P.13, When quoting works that showed more efficient repair in open chromatin it would be appropriate in addition to reference 66 to also quote: Adar S. et al, PNAS 2016 doi: 10.1073/pnas.1603388113. and Perera et al, Nature 2016 doi:10.1038/nature17437

4) I think the cell type the experiments were done in was not mentioned in the results and only in the materials and methods section.

5) The data produced should be openly accessible to the research community after publication – without having to request a code from the authors.

In summary, I think this is a novel and important paper, and after adequately addressing my concerns, it is well suited for publication in Nature communications.

Reviewer #3:

Remarks to the Author:

This is an interesting (mainly) functional genomics study of human cells lines recovering from a pulse of UV radiation. The authors present evidence that following a pulse of acute UV radiation, cells trigger a wave of transcriptional elongation on virtually all active genes and irrespective of their expression status. This is thought to be possible through a transcription elongation checkpoint override mechanism of the Promoter-proximal paused (PPP) RNAPII molecules. That way the elongating RNAPII could engage in transcription-coupled NER, which is thought to be more effective than global genome NER.

This is a very thorough study that leaves the impression it may have seen previous review/revision cycle(s). It consists of almost exclusively systems biological approaches and meta analyses and thus, the mechanistic insight is limited. However, this is norm for this kind of analysis and should not be understood as criticism.

I have only a couple of textual suggestions for revisions that I will briefly outline below. In summary, I can recommend this manuscript for publication in Nature Communications without hesitation.

Suggestion 1: I think the title of this manuscript is a bit misleading. Genotoxic stress is a very general term for stressors that induce DNA damage. However, only UV has been tested in this study. In addition, the response described in this paper would not make much sense for other types of DNA lesions that are not repaired by NER. I would therefore suggest to alter the title to: "A global unleash of transcription elongation waves in response to UV radiation restricts somatic

mutation rate".

Suggestion 2: I have a conceptual issue with the transcriptional response to UV that is described in this manuscript. I wonder why and how it may have evolved. UV light is a constant environmental genotoxic stressor on this planet and exposure to it does not occur in pulses but over several hours per day. The response described here however is a wave of transcriptional elongation that follows a pulse of UV, in other words, an environmental stressor that does not exist in nature. So it must have evolved for a different reason. I think this issue deserves a few sentences of discussion and I would be very interested to read the author's thoughts on this.

Finally, I noticed that there were quite a few typos in the manuscript and plenty of comma mistakes. I therefore suggest another careful proofreading and perhaps the use of a punctuation correction program.

Response to the Reviewers

We acknowledge the reviewers for their positive opinion on the relevance of our work. We believe that, by taking into consideration their constructive comments, the content and quality of the manuscript have greatly improved.

For Convenience, we also included a table summarizing the changes we made in the Figures and Supplementary Figures at the end of this letter.

Reviewer #1 (Remarks to the Author):

In this manuscript, Lavigne et al, examine the relationship between transcription and DNA repair after genotoxic stress (UV damage). The results are intriguing in that they see a shift in polymerase distribution after UC stress as polymerase at promoters become released into productive elongation. The shift is believed to be due to the release of P-TEFb from its repressive complex into a form that can activate paused polymerases. Although not shown here, UV-induced shift of P-TEFb has been shown previously. Using DRB to synchronize transcription elongation, they go on to show that the rate of transcription is slower after UV induction, and that the wave front of polymerase correlates well with location of TT-lesions (measured by CPDIP-seq in this study) and repair sites (measured by XR-seq in another study).

This suggests that the increased polymerase in the gene are responsible for recognizing lesions and stimulating repair mechanisms. Consistent with this, they go on to show that active genes in both lung and skin cancer samples have lower mutation rates in expressed genes. They suggest that the release of PPP results in protection for all active genes regardless of their original transcription levels. They are quite a few datasets here and the production and analysis looks to be of sufficient quality, for the most part. The conclusions seem to be well-supported by the data and they also make sense biologically. I recommend that the authors address the following issues before this work is considered for publication.

We thank the reviewer for his compliments and for his recommendation for publication. His comments and suggestions were very helpful, and they have been addressed. Appropriate corrections were included to improve the manuscript (see below).

Major points:

1) The Svejstrup lab recently published GRO-seq data after DRB synchronization and UV treatment: <http://dx.doi.org/10.1016/j.cell.2017.01.019> . They also show that transcription rate is reduced. The authors should be citing this article as well as addressing how their data compare.

We thank the Reviewer for pointing this out. Indeed, using a similar experimental approach to ours, Svejstrup's group has very recently found that the transcription elongation rate is globally decreased after UV irradiation. We are pleased to confirm that the elongation rates that they measured ('elongation rates of only 0.47 kb/min (10– 25 min) and 0.25 kb/min (25– 40 min)') correlate with the elongation rates we have calculated (New Figure 3c- 0.28 kb/min in average during the recovery period we investigated (10–60 min)). We note that this article was published shortly after we had finalised our initial version of the manuscript. We have

now included this reference in our revised manuscript and explain how well our data compares.

- Page 8, line 237: 'In addition, our model is consistent....'

2) They show that the wave front of polymerase correlates well with the positions of NER recognition and repair. To show a more causal role for transcription here, could the authors show the rate of recognition and/or repair if UV-stress is reduced in the absence of transcription elongation (i.e. in the presence of DRB?) They may already have this data.

We thank the Reviewer for giving us the opportunity to present a more advanced analysis on the relevant data: our DRB-ChIP-seq of RNAPII ser2P. We have emphasized the new analysis implications more explicitly in the revised manuscript (see below for explanations).

Without DRB inhibition, we have shown in old Fig. 3 that the average profile of NER-sensing by RNAPII (recognition of DNA lesions) and NER-repair (excision step, assessed by XR-seq) are enriched around TT loci for clusters located within the spread of the *de-novo* wave of transcription. In contrast, we have also shown in the same figures a lower, but detectable, recognition (sensing) and repair (excision) of TT loci that are positioned beyond the wave front.

Reviewer 1 is asking whether this difference would be cancelled in the case of inhibited *de novo* transcription elongation. As he pointed out, the data presented in Old Figure 2b (DRB-ChIP-seq of RNAPII-ser2P) could be analysed further to investigate whether the absence of *de novo* elongation can limit the boost in damage sensing. As we explained before, DRB inhibition eliminates the release of *de novo* RNAPII in gene bodies upon UV, because it blocks the transition of RNAPII into productive elongation. In accordance with the reviewer's suggestion, mapping RNAPII around TT loci when irradiation was performed in the presence of DRB could confirm if the advantages brought by the release of *de novo* elongating molecules (increased sensing of DNA damages) are lost. In other words, such result would confirm that the rate of recognition of UV-stress is reduced in the absence of *de novo* transcription elongation.

In line with a helpful comment of Reviewer 2 (see below) requesting that our analyses should be more quantitative, we have re-analysed our data. We have now included a precise quantification of the number of regions displaying high levels of RNAPII sensing and lesion excision in normal conditions (NO DRB, see New Figs. 4-5 and supplementary Fig. 9 and 10). More particularly, and of specific interest to be able to quantify the rate of damage recognition, we could count the difference in reads density at TT regions (Summit) and at the immediate nearby regions (Flanks) and defined an 'S-F score'. We find that in normal conditions (without DRB) for all the clusters falling within the range covered by *de novo* released transcription wave, there is a significantly increased proportion of TT regions with high 'S-F scores' for both RNAPII reads and XR-seq reads (Page 10, lines 273-95).

Interestingly, when analysing the DRB-RNAPII ChIP-seq data, we can now confirm that inhibition of *de novo* transcription elongation considerably limits the number of lesions recognised by RNAPII. In particular, as expected by the lack of wave release in these conditions, the difference in the number of lesions recognised (high S-F score) in the first clusters, which are normally spanned by the *de novo* wave without inhibition, is significantly flattened (New Supplementary Fig. 10). In other words, the difference in the number of regions showing stalled RNAPII at TT (High S-F score) between upstream and downstream TT clusters is significantly higher without DRB inhibition. Together, these results demonstrate that in the absence of a *de novo* boost of transcription (prevented by DRB), the rate of recognition of UV-stress is reduced to background levels. This implies that when drug is applied, cell's lesion recognition abilities match with the density of RNAPII already elongating prior to stress (Note: 'pri-elongating' is now used instead of 'pre-elongating' - see

below), and that cells lose the possibility to maximise lesion sensing/repair at lower expressed genes and at the most proximal part of the genes.

The manuscript (Page 11, lines 296-311) has been corrected accordingly.

3) How well do their transcription rate estimations match between ChIP-seq and nRNA-seq?

As also stated in our initial manuscript the transcription rates that we estimated by nRNA-seq and ChIP-seq match nicely: “UV-triggered wave propagation rate is consistent between our ChIP-seq (Supplementary Fig. 3i) and nascent RNA-seq (Fig. 2e) analyses” (page 11).

When we compare the two time-course experiments, they do correlate nicely ($R^2=0.89$). This is now mentioned in the manuscript.

- Page , lines 219-22:

‘In fact, transcription elongation rates of the de novo released complexes decreased during recovery, in a similar way to pri-elongating RNAPII (see Figs. 2c and 3c), confirming the changes in elongation rates measured overall ($R^2= 0.89$, compare Fig. 3c with Supplementary Fig. 3i).’

4) There is a clear UV-dependent increase in transcription in the genomic data, however, in cells (Figure S4b), there is a reduction at 2hr. Please explain.

We apologize that we did not clearly explain in the previous version of the manuscript this UV-induced changes in nascent RNA. To put stronger emphasis on the overall drop of nRNA-seq signal upon UV, we have included a more distinct labelling in New Supplementary Figure 4a, and highlight the regions of the gene body enriched in signal of nRNA after UV versus the ones depleted. In this way the reader can appreciate better that the overall nRNA-seq signal is significantly reduced at 2 h because log₂ FC are negative for the majority of the gene length and therefore also in average for the whole gene length. This result is in agreement with the overall depletion in EU-fluorescence presented in panel b of the same figure.

We have now re-formulated the text as follows.

Page 6, lines 154-160:

‘In turn, RNAPII is promptly released into productive elongation (ser2P) and regions directly downstream of PPP display a substantial increase in nascent RNAs (nRNAs) between 0,5 and 2 h post-UV (Supplementary Fig. 4a). We also identified an important depletion of RNA synthesis in more distal regions of the gene bodies, and confirmed a global decrease of RNA production in the nuclei 2 h after UV-irradiation (Supplementary Fig. 4a and b, respectively).’

5) Page 7: “Interestingly however, when cells were exposed to UV, nascent transcription recovery could still be detected to a significant degree, suggesting that UV damage does not prevent de novo RNA synthesis.”

Comment: As mentioned in point 1, this observation has been reported, and should be addressed here.

As explained above, we have now referred to the very recent work from Svejstrup's lab. In their set-up, Williamson et al. interpret the incorporation of Br-UTP after UV to represent active RNAPII molecules and '*suggest that initiation and transcript elongation in the promoter-proximal areas still occurred, while progress further into genes was very slow or prohibited*'. We now describe in the text that the results we obtained relied on both nascent RNA and RNAPII ChIP-seq methods after DRB-release and we discuss how our findings in general corroborate and extend their observations.

- Page 8, lines 209-15: 'Surprisingly, UV irradiation did not abolish the recovery of nascent transcription after DRB removal, as previously suggested⁴⁸. .../... Therefore, we suggest that UV irradiation is by no means sufficient to prevent release of RNAPII into productive elongation.'

- Page 9, lines 237-9: 'In addition, our model is consistent with and explains recent findings suggesting that transcription initiation and elongation still occurs in promoter-proximal areas upon UV irradiation although progress into gene bodies is very slow⁴⁸'.

Minor points:

1) *Figure 5 and 6 seem redundant and could possibly be condensed and joined if space is of concern.*

We thank the Reviewer for his suggestion. For better comprehensibility and to comply with a related comment of Reviewer 2 (see below), we have now reorganised those figures, along with the associated old supplementary Figure 10.

A unique main Figure (New Fig. 6) and the new Supplementary Fig. 12 clearly show now the comparable low and uniform mutation rates, which were detected in all expressed genes in two different tumour genomes. As explained in the modified text and methods sections, we have corrected our analysis for exon densities when needed (Reviewer 2 request), and we also show a more detailed analysis depicting mutation prevalences as a function of gene expression levels. We estimated gene expression in relevant cells using nRNA-seq data rather than mRNA-seq data. These modifications allowed us to put stronger emphasis on the wide clinical importance of the newly identified mechanism of wave release, which limits mutagenesis in tissues exposed to various relevant genotoxic stresses. We conclude that 'We uncovered low and particularly homogenous level of genetic changes in all active genes, in both analysed genomes.../... independently of the exact cause of DNA damage.'

Page 14, lines 397-405

2) *For clarity on the advantage the described mechanisms confer on active genes, they could incorporate what happens to no-active genes into model in figure 7.*

We fully agree with the Reviewer. We have now included non-active genes in the model (New Fig. 7). This allowed us also to describe further the cases when NER fails or it is not recruited efficiently during stress recovery, as unrepaired lesions may provoke error-prone DNA synthesis and result in mutations during replication.

3) Bottom of page 9. Use of 'pre-elongating' is confusing. Consider rewording.

Although we had defined what we meant by 'pre-elongating' in our context, we agree that this term was confusing. We believe that a more suitable term to define the polymerase already elongating prior to stress could be 'pri-elongating'. We have now redefined this subset of polymerases as 'pri-elongating', and accordingly we have replaced this term through the manuscript.

Page 7, lines 196

4) Page 10: "A prediction from our findings would be that, the mutation rate of NER associated signatures should remain low in all active genes independently of their expression level."

Comment: No comma necessary

We thank the reviewer, the correction has been included (Page 12, line 360-1).

Reviewer #2 (Remarks to the Author):

The authors unveil a global release of paused polymerase in response to UV treatment. This phenomenon is extremely interesting and as the authors suggest, could assist in detection of DNA damages and reduce mutagenesis.

The described findings are novel and of high importance to people coming from a broad range of fields – including of course researchers in the repair and transcription communities. Following the different isoforms of PolIII after UV is an elegant and complicated experiment – and I find the comprehensive characterization of PolIII occupancy together with nascent RNA synthesis convincing and of great importance.

The majority of my comments relate not so much to the quality of the data but to it's interpretation.

We appreciate the Reviewer's positive and stimulating remarks on the relevance and quality of our work. We found some very helpful comments and suggestions for clarifying the interpretation of our data and included new analyses in the revised version of our manuscript.

Major comments.

1) To what degree are the experiments quantitative?

The finding that RNA PolIII release from promoters in an elongation wave is very convincing. In general, however, I find the more “quantitative” claims the authors make to be less convincing.

Comparing different ChIP-seq data sets in a quantitative manner is problematic. The authors rely on Chromatin fraction- and ChIP-western blots to show the overall changes in the abundance of different isoforms – but to make these western blots truly convincing I would like to at least see quantification (and in replicates).

We analysed in detail the ChIP-seq data of the three isoforms of RNAPII (Old Fig. 1 and Supplementary Figs. 1, 2 and 3 a-h). In particular, we have developed a script to count the exact number of ChIP-seq reads overlapping with promoters or with gene body regions for each gene (see below). We thus believe that the quantitative aspects of the changes in RNAPII isoforms localisation and abundance around promoter-proximal regions are extensively addressed.

Yet, we agree with the Reviewer that the western blots shown in old Supplementary Fig. 3 j, k were only qualitative. We had initially conducted these experiments to support the novel genomic analyses provided in Figure 1 and Supplementary Figures 2 and 3 a-h and also to confirm previously reported biochemical evidence.

To be more accurate, we now have followed the Reviewer’s suggestion and we have conducted new chromatin fractionation analyses by western blot in duplicates. We quantified bands intensity in all isoforms and normalised to NO UV levels (steady-state). In the revised manuscript (New Supplementary Fig. 3j, k), we also indicate the average percentages \pm SEM (calculated for the replicated experiments) and confirm in a more quantitative manner the significant losses of RNAPII-hypo from chromatin (\log_2 FC \sim -2, represents a loss of 75%). In fact, we note that this result match nicely with our ChIP-seq data, from which we could estimate losses of RNAPII-hypo reads density at peak summits (promoter) to be about 75% too (Supplementary Figure 3c).

Page 6, lines 144-51: ‘As expected, the levels of pre-initiating polymerases (RNAPII-hypo) were significantly reduced 2 h post-UV (\log_2 FC \sim -2, which represents a loss of about 75%)....’

The escape indices of the polymerase is defined as a ratio between polymerase at the gene body and the promoter. The changes in escape indexes can be a combination of switching from paused to elongating, but can also be a cause of differential dissociation from DNA.

The Reviewer is totally right about the fact that theoretically, a change in EI ratio could reflect differential dissociation of RNAPII. However, the dissociation of RNAPII from DNA

at promoters is probably limited, because a supply in newly-released molecules from promoter regions is necessary to feed the *de novo* wave of elongation described in our model.

Relying on a precise quantification of the reads distributions for each isoform in both promoter and gene regions separately, we do show an increase in reads density (Rd) of elongating polymerase in the gene body, which is compatible with a more significant shift of RNAPII-ser2P reads into gene bodies (New Supplementary Figs. 2b and 3e-g). In parallel, we also report in the text “a clear increase of nascent RNAs at 5’ ends of the genes between 0,5 and 2 h post-UV (New Supplementary Fig. 4a).” As explained above in response to a comment of Reviewer 1, increased incorporation of labelled 5’EU could theoretically be an effect of increased RNA polymerisation rate by a constant or decreased number of elongating molecules. This idea would be compatible with the reviewer suggestion that changes in EI ‘*can also be a cause of differential dissociation from DNA*’. Nonetheless, we explain in the new version of the manuscript that nascent RNA signal is most likely a consequence of the increased number of RNAPII molecules synthesizing more RNA molecules (at an equal or decreased rate).

Thus, together, our results confirm that RNAPII dissociation from DNA could not explain the combined increase of RNAPII and nRNA signal observed at the beginning of the genes during early recovery phase. We are thus convinced that EI increase is due to a switch from paused to elongating RNAPII. To clarify the interpretation of our data, we have changed the text accordingly.

- Page 5-6 , lines 126-134 and lines 151-7

The authors claim that EI for low expressed (highly paused) genes increase more than EI for highly expressed. If this is the case, and there is indeed an “override” of the transcriptional program – You would expect the RNA PolIII levels after UV to NOT correlate with the previous steady state transcription levels. Is this the case?

Yes, this is actually the case. We did state in the manuscript that “RNAPII EIs of normally highly-paused/low-escaping genes (low expression levels, PCC = 0.3711) are more significantly increased than EIs of less paused (highly expressed) genes (Fig. 1f).” This quantification was performed to confirm the fact that upon UV, the density of elongating RNAPII in the first kb of gene bodies is more constant across the whole range of gene expression (Old and New Figure 1d). This observation is in sharp contrast with the situation we observed in the steady-state prior to UV irradiation where the level of RNAPII in gene bodies follows the gradient of EI and thus RNA expression levels.

To be clearer, we reformulated the text as such:

- page 7, lines 173-7) ‘we demonstrate that EIs of normally highly-paused/low-escaping genes are more significantly increased than EIs of less paused (highly expressed) genes (Fig. 1f). In other words, RNA PolIII levels in gene bodies after UV do not compare with the steady state levels (more uniform yellow signal in Fig. 1d upon UV)...’

And a more general note – this entire mechanism is still limited only to those genes that were expressed to begin with – and so not a complete override.

We apologize for not explaining in a clearer way what we meant. We would like to clarify that ‘overriding’ is not referring to the number of genes expressed, but to the frequency of firing RNAPII in productive elongation. Hence, ‘overriding’ qualifies the changes in the level of expression of the constitutive expressed genes. We have corrected our text to be less confusing to the reader: Page 4, lines 84-6: ‘transiently adjusts the regular transcription firing program of active genes in human cells to accelerate damage sensing and to actively preserve genetic integrity.’

2) *Do polymerases fall off or continue after encountering the damage?*

A longstanding question in transcription-coupled repair has been whether RNA polymerases that encounter damage dissociate or continue after repair. The recent paper by Andrade-Lima et al had brought forth evidence for dissociation. Here, the authors performed a very sophisticated experiment in which they used DRB to eliminate actively elongating RNA polymerases from genes and follow transcription of the de-novo released polymerases by RNA PolII-Ser2 ChIP. These experiments indicate that after release, these RNA polymerases continue throughout the genes and do not dissociate.

The Reviewer is right in the sense that our data challenges conclusions presented in Andrade-lima *et al.*. Indeed, both our experimental set-ups using DRB to inhibit transition into productive elongation indicated that whether we focus on de novo released RNAPII molecules (New Fig. 3) or pri-elongating molecules (new Fig. 2), we see progression of the waves of synthesizing RNAPII detected by ChIP-seq and nRNA-seq. Even at late time points of recovery (+2h and +6h) we identify wave progress towards the most distal part of those genes (see Fig. 1e) in our ChIP-seq experiments. Corroboratively, RNA synthesis is also detected at +2h with or without DRB even at distal regions from TSS for long genes (>60 kb) (New Fig. 3 and Supplementary Fig. 4). We have now explained in the manuscript that:

- Page 9, lines 234-9: ‘Together, our results challenge recent conclusions^s. We argue that differences in nascent RNA read densities between the proximal and distal parts of the genes are more likely to be the result of an increase in RNA synthesis in the proximal region rather than a total loss of RNAPII molecules at the more distal regions. In addition, our model is consistent with and explains recent findings suggesting that transcription initiation and elongation still occurs in promoter-proximal areas upon UV irradiation although progress into gene bodies is very slow⁴⁸.’

The 2015 paper followed nascent RNA sequencing and not RNA polymerase occupancy after UV. Since Andrade-Lima et al saw lower levels of coverage at 3’ ends of genes, their conclusions were that RNA polymerases fall off as they encounter damages. After DRB conditions, Lavigne et al also follow nascent RNA transcription – and claim that since transcription does recover it indicates UV damages do not prevent de novo synthesis.

This part of the interpretation is less clear to me. It does appear that transcription is lower at the 3’ regions of the genes – even at 2h where at 1h transcription was evenly distributed without damage (Figure 2e,f). The authors explain this by the slower progression of polymerases over a damaged template. The prediction would be that at later time points, as the polymerase progress, RNA synthesis at the 3’ end is recovered – bust such a time point is not shown.

The reviewer may have missed it but we had already shown in Supplementary Fig. 4a that after 24h nascent RNA levels are reset to normal (black line). This is in agreement with the fact that, for RNAPII-ChIP-seq, we also see a reset of the distribution at + 48h and even with stronger doses of UV (supplementary Fig. 7). In the text we mention:

- Page 12, lines 351-3: ‘this phenomenon contributed to the eventual reset of steady-state RNA levels and RNAPII distributions at later time points (see Fig. 1d, e, and Supplementary Figs. 2d, 3h, j, k, and 4a, b, 7a, b)’”.

Our Chip-seq data presents such an intermediate view on transcription wave progression at UV +6h (New Fig. 1 d-e, in turquoise). This progression is compatible with the fact that pri-elongating polymerases keep on detecting lesions downstream of the wave during this whole period, which suggest that a significant amount of these polymerase are still bound to chromatin even at +6h in distal regions of long genes (New Fig. 4d, e). In addition, the analysis of RNAPII distribution over TT loci also confirms distal TT regions show much less stalling of RNAPII at +6h than at +2h (New Supplementary Fig. 9d). In accordance with the comments below suggesting to quantify better the stalling of RNAPII, we have now reformulated the text as follows:

- Page 12, lines 351-3: ‘At later time points during recovery, as more lesions have been repaired, the balance between lesion-stalled and elongating polymerases was progressively inverted (compare 2h and 6 h in Supplementary Fig. 9d)’

Finally, we note that in the data presented in the Andrade-Lima et al. paper, authors also analysed nRNA-seq signal at +6h. In accord with our analysis, their data support the idea that RNAPII keeps on progressing in gene bodies between +2h and +6h and this process is accompanied by a regain in nascent RNA synthesis at regions situated in the distal parts of the long genes.

3) How does the post-UV transcription relate to the underlying damages and their repair?

The authors performed experiments to map CPDs following UV in a genome-wide manner using CPDIP-seq. These experiments are not thoroughly validated and truthfully it is unclear if they were successful. However, previous works to map CPDs genome wide (authors should also quote Teng Y et al 2011, doi: 10.1093/nar/gkq1036) have shown that generally, damage profiles are dictated by the TT frequency- and so the approach the authors took of simply following patterns centered on TT nucleotides is valid and acceptable.

We apologise for not providing all the relevant data for our CPDIP-seq experiment. We have now included more details and show that the pull down of CPD-enriched DNA was successful with over 16-folds enrichment in CPDs in the pulled down DNA compared to Input DNA (New Supplementary Fig. 8c, step 5). We also provide the read density profile of CPDIP-seq reads for genes longer than 60 kb to clarify that although apparently low, we can detect significant amounts of CPDs and their distribution is uniform along expressed genes, regardless of genes’ EI (New Supplementary Fig. 8c, step 7). We thank the reviewer for pointing out the yeast analysis also suggesting that damage profiles are dictated by the TT frequency, and we have now included this point in the manuscript to further validate our experimental approach.

The authors compared the pattern of de novo elongating RNA polymerases to both TT density and the patterns of repair measured by XR-seq. However, wouldn't a similar correlation be expected to total RNA polymerase density, or the steady state levels of RNA polymerase prior

to UV? The quantitative nature of the result is not clear. What proportion of the damages are recognized by these de novo released polymerases and how many were recognized by the polymerases that were actively elongating at the time of UV? Perhaps an experiment comparing repair after DRB vs in non-DRB treated cells could delineate the contribution of these newly released polymerases?

The reviewer is right about the possibility that the underlying RNA polymerase density impacts on the probability to find RNAPII near a TT region. However, we had stated in the text that: ‘We found that co-localization of RNAPII with potential adduct sites was maximal in categories located upstream of the wave front position at 2 h post-irradiation (Fig 3a,b: clusters I-III)’ and ‘Co-localization pattern of RNAPII was not observed at other control genic loci (i.e. intron-exon junctions) (data not shown)’.

The confusion could come from our lack of adequate explanations of the results. On the density profiles (Old Fig. 3b and New Fig. 4b), we interpret the increase in signal near the centre of the region to be specific to RNAPII stalling at TT loci, while the flanks of the regions analysed represent the background levels of RNAPII passing by without necessarily stalling. Therefore, without being quantitative, we concluded that the signal around the centre is higher than the background detected at the flanks, especially for +2h and for clusters upstream of wave front positions and reported this as ‘co-localisation’.

In fact, the data also shows (although we had not directly commented on it) that in comparison, the signal observed for NO UV (in blue), which represents the steady state levels of RNA polymerase around TT loci prior to UV, is not increased at the centre of the regions. In other words, total RNA polymerase density is not specifically enriched at TTs before UV, and the levels of RNAPII around TT is almost equal regardless the position of TT in the gene body. Finally, we had mentioned as ‘data not shown’ that we also verified that RNAPII profiles detected around exon start regions (a control region set that is not preferentially enriched for dipyrimidines) do not show specific increase of RNAPII reads density around the centre of the region.

Overall, we agree that the quantitative nature of these results could be improved and shortcuts in the interpretation of the original Figure could be more thoroughly explained to clarify the importance and novelty of the results.

Consequently, we have thoroughly re-analysed the primary data by reconstructing a whole bioinformatics pipeline and we updated the manuscript accordingly. We quantitatively determine: i-the level of specific-stalling vs background level of RNAPII at both TT loci and exon start regions, and ii-the number of regions displaying high levels of RNAPII sensing and of lesion excision in response to UV (see New Figs. 4-5, and New Supplementary Fig. 9). More particularly, we indicate in the text and Methods that we counted the difference in reads density at TT regions (Summit) and at the immediate nearby regions (Flanks) to define a ‘S-F score’ representative of the specific vs background level.

We find that, for all the clusters falling within the range covered by *de novo* released transcription wave, there is a significantly increased proportion of TT regions with high ‘S-F scores’ for both RNAPII reads and XR-seq reads. This result suggests that more damages are recognized and excised by *de novo* released polymerases than by pri-elongating molecules.

Importantly, applying this quantitative approach to control regions that do not contain dipyrimidines (i.e. exon start sites) (New Supplementary Fig. 9a-e) clearly confirms that the signal of RNAPII observed at the centre of the TT regions is absent around exon start regions, meaning that specific accumulation of RNAPII around a given region (co-localization) depends on the likeliness of these regions to be damaged. This Figure also quantifies the underlying proportion of regions enriched for passing-by RNAPII (Exon Start top table in New Supplementary Figure 9d) due to the passage of the *de novo* released wave. We verified

that the percentage of TT regions with high FC of RNAPII signal is higher than that of exon starts regions (New Supplementary Fig. 9e). This difference was more significant, especially where we recorded more specific stalling of RNAPII (percentage of regions with high S-F score) (compare chi square test values in New Supplementary Fig. 9e with lower tables in New Supplementary Fig. 9d).

In accord, the manuscript has been significantly rewritten:

- Page 10, lines 273-95

Interestingly, when analysing the DRB-RNAPII ChiP-seq data in this way, as also suggested by Reviewer 1 (see above), we confirm that inhibition of *de novo* transcription elongation considerably limits the number of lesions recognised by RNAPII. In fact, the difference in the number of regions showing stalled RNAPII at TT (High S-F score) between upstream and downstream TT clusters is significantly higher without DRB inhibition (New Supplementary Fig. 10). Incidentally, this new analysis is of significant importance as it enabled us to establish the proportion of lesions recognized solely by the polymerases that were actively elongating at the time of UV as requested by the reviewer.

- Page 11, lines 296-301

Thus, as previously stated from our qualitative analysis, we conclude that RNAPII stalling is increased at 2h especially for clusters located upstream of the wave front. Nonetheless, our new quantitative analysis also demonstrates that in the absence of a *de novo* boost of transcription, the rate of recognition of UV-stress is reduced to levels that depend on the density of pri-elongating RNAPII at the time of stress. This means that without de-novo wave release cells lose the possibility to maximise lesion sensing/repair at lower expressed genes and at the most proximal part of the genes. We thus delineate better the contribution of these newly released polymerases in maximizing DNA repair.

- Page 11, lines 301-311

In the interpretation of XR-seq data, the authors should bear in mind that this data too is not quantitative. In wild type cells, while very strong XR-seq signal is observed at 1h – The majority of the lesions are still there– and for hours after. It is problematic to assess the quantitative differences in overall repair levels between WT/XPC/CSB cells from the XR-seq patterns. Furthermore, XR-seq does not measure total repair level, but only the repair level at the 10-30min leading to that time point. Therefore it is not at all clear that the wave of RNA polymerization is clearing up damages or contributing significantly to the recovery from damage.

We have also applied now in New Figure 5 the quantification of specific XR-seq signal around the center of the regions (New Fig. 5c, e) and we are able to quantify the number of regions that display a high S-F scores (New Fig. 5d). In fact, this quantification shows that in cells capable of recruiting TC-NER (WT and XP-C cells), the level of repair is higher upstream of the expected RNAPII wave front calculated at the same time point of recovery (New Fig. 5 e).

- Page 11, Lines 319-25: ‘we discovered that excision activity is enriched for TT loci located close to TSS (Fig. 5 a,b, clusters I-II), as confirmed by the high ‘S-F scores’ (Fig. 5 c, clusters I-II) and the increased proportion of regions undergoing repair (Fig. 5d, upstream of the wave front). In accord with the functional advantages offered by the release of a *de novo* transcription wave, which facilitates lesions sensing (see Fig. 4), the repair of TT loci located

downstream of the wave front was lower than for the upstream regions, especially in TC-NER-proficient cells (Fig. 5 c-e clusters I-II (up) vs III-VI (down)).

We certainly agree with the reviewer that XR-seq does not measure the total repair level, but only the repair level at the 10-30min leading to that time point, and that most certainly, a majority of the lesions are still there unrepaired. Nonetheless, our point was not to show that the repair was completed at this time point and we only claim that more loci are repaired under the theoretical span of the newly identified *de novo* wave of RNAPII. In fact, the percentage of regions we calculated to be excised are quite low and we can estimate that in WT cells, for instance, only 2.4% of the regions of the cluster I show significant excision signal (New Fig. 5d). It could be that the rest of the regions have already been excised and the signal cannot be recorded at this time point. But most likely, considering that RNAPII stalling increases up to + 2h and regions of the cluster I still get recognized after + 6h (New Fig. 4), we can speculate that most of the TT loci have not yet been excised. We changed the text accordingly as such:

- Page 12, Lines 344-46: 'The XR-seq method detects lesions being excised at the time of the assay³⁸. In line with this, we found that only a small proportion of regions were being repaired at +1 h, which suggests that completion of the whole repair process may take several hours

Our results now support the idea that the *de novo* wave of RNA polymerization facilitates clearing up the damages and indirectly this mechanism does contribute to maximize recovery from damage.

- Page 12, Lines 344-46: 'We therefore conclude that, as an important part of the process underlying the response to stress, *de novo* wave release of elongating RNAPII in active genes promotes both an accelerated and a more uniform removal of transcription-blocking DNA lesions by TC-NER activity on the TS, but also because of facilitated GG-NER on the NTS. This mechanism therefore contributes to maximize recovery from DNA damage.'

The authors show that transcription coupled repair occurs equally well on all expressed genes despite differences in the escape indexes (Fig 4a). However, it was previously shown that transcription coupled repair correlates to the RNA levels from the genes prior to UV treatment (Hu et al, 2015). This highlights the importance in incorporating the initial transcription levels in the analysis.

Indeed, our results could appear different than the previous analysis performed by Hu et al.. Methodological and conceptual adjustments to their approach allowed us to refine the ranking of gene expression. We are based on actual RNAPII densities rather than processed mRNA levels. Our supplementary Fig. 1 explains how we established the list of actively vs inactively transcribing genes in NO UV steady state conditions. Genes enriched for RNAPII ChIP-seq reads of either isoform were analysed to establish a robust list of active genes. We also verified that density of RNAPII-ser2P around promoters (Dp) correlated better with nRNA-seq data (PCC= 0.78) than mRNA-seq data (PCC= 0.71) as expected from possible effects of post-transcriptional regulation or from increased noise in the latter dataset. In the manuscript we invite the reader to check Supplementary Fig. 1 for details:

- Page 5, lines 101-7

Analysing RNAPII-ser2P EI allowed us to stratify genes according to the probability of firing elongating molecules in gene body and we also verified that expression levels (nRNA-seq) were reasonably correlated to EI (PCC = 0.3711). In fact, Fig. 1d reflects nicely how RNAPII

density in gene bodies in steady-state NO UV is decreasing with EI. To be consistent with the analysis of RNAPII showing that RNA PolII levels after UV do not correlate with the previous steady state transcription levels (see above) we asked how the one genes' EI impacts on its probability to be repaired. We now present the results in Supplementary Fig. 11 and demonstrate that for both cell lines proficient for TC-NER, in a representative set of long genes (>60 kb), XR-seq activity is uniformly boosted in regions falling under the span of the de novo released wave at +1h (i.e. TSS to +32.5 kb, see Fig. 1e) regardless of EI. Thus we modified the manuscript to be clearer on the interpretation of our results:

- Page 11, lines 326-8: 'In TC-NER proficient cells, we also found that excision of DNA lesions on the template strand (TS) was generally amplified in active genes compared to inactive genes, with a prominent signal at the 5' end (Supplementary Fig. 11).'

4) *How does the phenomena of de novo release of elongating RNA polymerase relate to mutagenesis?*

In their analysis, the authors calculated the frequencies of cancer-linked mutations on the transcribed and non-transcribed strands of genes. I have two major reservations about their analysis.

First: The mutations analyzed were C>T - while the damages that were discussed all along were TT dimers.

As stated in the main text: ' We focused on two datasets⁸ previously linked to NER activity, as they display "T-class"(Transcriptional asymmetry) -associated base substitutions^{8,14}. More particularly, skin melanoma (Fig. 5) and lung adenocarcinoma tumors (Fig. 6), are known to show strong probabilities for UV (C>T) and smoke (G>T) induced mutagenesis respectively.' The papers we referenced explain extensively how C>T and the G>T substitutions have been demonstrated to be the most representative signatures of UV and smoke unrepaired damages, respectively. Old Supplementary Fig. 10 and Methods section was also designed to explain our rationale for using C>T and G>T mutations as a readout of impaired NER activity. For space constraints, we had specified these details in the 'Mutation rates analysis' section of the Methods: "Noteworthy, CC, CT and TC CPDs, although less frequent than TT dimers⁶¹, generate more frequently error-prone translesion DNA synthesis, which results in more C > T mutational events^{9,102,103,68}. For smoking-related impairments, G > T (or the reverse complement C > A) substitutions were selected because they correspond to the most abundant smoking adducts-generated mutations repaired by TC-NER^{8,10,14}. In order to extract the trinucleotide context of these selected mutations, we retrieved from hg19 fasta file the flanking bases around each selected mutation. We then filtered all the hits corresponding to the most frequently characterized trinucleotide events for UV (T(C)C > T(T)C (and G(G)A > G(A)A)) and smoking (T(G)G > T(T)G (and C(C)A > C(A)A)) in Melanoma and Lung Adeno datasets respectively and generated BED files." We have now included the relevant explanation in the main text:

- Page 13, lines 366-77

Second: The Alexandrov et al. mutation data set was produced primarily by exome sequencing. Therefore, in the analysis only exons should be taken into account and not the entire transcript.

We thank the reviewer for pointing out this inaccuracy. We explain in New Fig. 12 b,c what samples were used in the analysis. We explain that whole genome sequencing (WGS) data available only for lung adenocarcinoma was preferred to whole exome sequencing (WES) as it provides insights in introns sequences. Since only WES data was available for melanoma samples, we corrected the calculated prevalence of mutations as a function of the underlying density of exon that we established in each gene. Re-analysis also included a change in the data we used to stratify mutation prevalence, as we used publically available data for nascent RNA available for two representative cell lines (human skin and lung fibroblasts) (Supplementary Fig. 12d,f and Methods). In this way, we were able to demonstrate in a more robust manner that mutation rates remain low and uniformly distributed along gene bodies, and are not significantly dependent on steady-state levels of RNA synthesis.

- Page 13, lines 380-92

Lastly, and likely beyond the scope of this paper: Do the authors have any indication if this phenomena is specific to UV or occurs in response to other DNA damages or even other stress types?

As explained above, we do not only associate *de novo* wave release to more efficient repair of CPDs after UV in all expressed gene in irradiated cells, but also to the homogenous repair of both UV lesions and smoke cigarette adducts across all expressed genes in exposed tissues, thus limiting the mutation rates in all transcribed genes regardless of nRNA production rate, which is intimately correlated to RNAPII density and EI.

Minor comments:

1) Showing that DNA from RNA PolIII ChIP is enriched in DNA damage 2h after UV: The dot blot shown in Fig 8e does not contain a negative control blot, and I understand blotting 200pg of damaged DNA is difficult. I would appreciate including the control (undamaged) DNA in the blot, as well as adding error bars to the quantification.

To be more accurate and verify once more our results, we repeated this analysis. The new Supplementary Fig. 9f includes blots of NO UV (undamaged control) samples, which show the background levels detected in our experimental set-up. We present blots for 2 biological replicates of inputs, and for 3 ChIPs series. We explain in Methods that we quantify the normal average FC between ChIP and Input at each time point by removing background signal calculated in the NO UV, and we estimate the s.e.m. across the 2 biological replicates (error bars). In this way, we find that the biochemical approach corroborates nicely the novel and quantitative genomic approach (see Supplementary Fig.9 d,e) showing that indeed RNA PolIII ChIP is enriched in on CPD 2h after UV (see Supplementary Fig. 9d; changes in the number of regions with high S-F scores)

2) Please make figure labels a bit larger. Figure 2C was impossible to read in print.

Aware that Fig. 2 was too packed, we have decided to split Old Fig. 2 in 2 different Figures (new Figs. 2, 3) and we enlarged the fonts to be more readable.

3) P.13, When quoting works that showed more efficient repair in open chromatin it would be appropriate in addition to reference 66 to also quote: Adar S. et al, PNAS 2016 doi: 10.1073/pnas.1603388113. and Perera et al, Nature 2016 doi:10.1038/nature17437

We have now included these references in the revised manuscript.

4) I think the cell type the experiments were done in was not mentioned in the results and only in the materials and methods section.

We have now added the information in the beginning of the Results section too.

5) The data produced should be openly accessible to the research community after publication – without having to request a code from the authors.

All raw sequencing data generated for this manuscript have been deposited in the GEO database and will be released once the manuscript is accepted for publication.

In summary, I think this is a novel and important paper, and after adequately addressing my concerns, it is well suited for publication in Nature communications.

We thank the reviewer for acknowledging the novelty and importance of our work.

Reviewer #3 (Remarks to the Author):

This is an interesting (mainly) functional genomics study of human cell lines recovering from a pulse of UV radiation. The authors present evidence that following a pulse of acute UV radiation, cells trigger a wave of transcriptional elongation on virtually all active genes and irrespective of their expression status. This is thought to be possible through a transcription elongation checkpoint override mechanism of the Promoter-proximal paused (PPP) RNAPII molecules. That way the elongating RNAPII could engage in transcription-coupled NER, which is thought to be more effective than global genome NER.

This is a very thorough study that leaves the impression it may have seen previous review/revision cycle(s). It consists of almost exclusively systems biological approaches and meta analyses and thus, the mechanistic insight is limited. However, this is norm for this kind of analysis and should not be understood as criticism.

I have only a couple of textual suggestions for revisions that I will briefly outline below. In summary, I can recommend this manuscript for publication in Nature Communications without hesitation.

We thank the reviewer for his positive comments and for recommending our manuscript for publication.

Suggestion 1: I think the title of this manuscript is a bit misleading. Genotoxic stress is a very general term for stressors that induce DNA damage. However, only UV has been tested in this study. In addition, the response described in this paper would not make much sense for other types of DNA lesions that are not repaired by NER. I would therefore suggest to alter the title to: "A global unleash of transcription elongation waves in response to UV radiation restricts somatic mutation rate".

In our final analysis, we had already extended our conclusions to another very common and important environmental stress, which consists of cigarette smoke. For this reason, we believe that our title is valid. We do not only associate *de novo* wave release to more efficient repair of CPDs after UV, but we also provide insights on how such mechanism fits with the homogenous repair of both UV lesions and smoke cigarette adducts across all expressed genes in tissues exposed to various genotoxic stresses.

Suggestion 2: I have a conceptual issue with the transcriptional response to UV that is described in this manuscript. I wonder why and how it may have evolved. UV light is a constant environmental genotoxic stressor on this planet and exposure to it does not occur in pulses but over several hours per day. The response described here however is a wave of transcriptional elongation that follows a pulse of UV, in other words, an environmental stressor that does not exist in nature. So it must have evolved for a different reason. I think this issue deserves a few sentences of discussion and I would be very interested to read the author's thoughts on this.

Many researchers in the DNA repair field are fully aware that UV light is a constant environmental genotoxic stressor on this planet. However, it is a common practice in the lab to estimate that the dose of UV exposure is applied in brief pulses of concentrated light rather than prolonged exposure of diffuse light. In real life, however accumulated exposure to UV light can also provoke overcoming a signalling threshold at which the response we present here can be activated. It can be speculated that during its lifetime, a single cell is able to determine when dangerous levels of exposure have been reached to justify a check-up. One can anticipate that cells could reset their sensor after each episode of repair. We also anticipate that the levels of exposure requiring the activation of the 'safe-mode' would certainly be lower for cells that replicate often, as these will need to benefit from a fast round of repair before they can divide safely. We have included a paragraph along these points in the discussion:

- Page 17, lines 495-501

Finally, I noticed that there were quite a few typos in the manuscript and plenty of comma mistakes. I therefore suggest another careful proofreading and perhaps the use of a punctuation correction program.

We have proceeded with careful proofreading of the new manuscript and have used a punctuation correction program to improve its quality.

OLD FIGURE	NEW FIGURE	Comments
1	1	NO Change
2 a-c	2 a-c	Numbering change and as requested by reviewer 1 -Larger size -Included elongation rates values on the bar chart.
2 d-f	3 a-c	Numbering change and as requested by reviewer 1 -Larger size -Included elongation rates values on the bar chart.
3 a,b	4 a,b	Numbering change In new Fig. 4b, added a schematic for better comprehension of analysis.
N/A	4 c-e	New panels: Quantitative analysis as requested by reviewer 2.
3 c,d	5 a,b	Numbering change In new Fig. 5b, added a schematic for better comprehension of analysis.
N/A	5 c-e	New panels: Quantitative analysis as requested by reviewer 2.
4	6	Numbering change
5	7 a-c	Numbering change and Re-analysed data As requested by reviewer 1, old Fig 5 and 6 were condensed in one Figure. As requested by reviewer 2, mutation data were analysed separately for whole exome and whole genome, normalised to number of samples and exon densities if needed.
6	7 d-f	Numbering change and Re-analysed data As requested by reviewer 1, old Fig 5 and 6 were condensed in one Figure. As requested by reviewer 2, mutation data were analysed separately for whole exome and whole genome, normalised to number of samples and exon densities if needed.
7	8	Numbering change Model updated for clarity reasons and in accordance to comments by Reviewer 1.
S1	S1	NO Change
S2	S2	NO Change
S3	S3	New western blots figures in j and k, with quantification bar charts as requested by reviewer 2
S4	S4	NO Change
S5	S5	NO Change
S6	S6	NO Change
S7	S7	NO Change
S8 a-d	S8 a-d	Panel c was updated with new analysis of CPDIP-seq data, as recommended by Reviewer 2 for better clarity. We highlighted that the pulled-down DNA used for library preparation is enriched more than 16-fold for CPD damages, and we included an average profile plot showing the even distribution of CPD damages over gene bodies
N/A	S9 a-d	New Figure including a control analysis of the distribution of RNAPII-ser2P reads around exon start regions that are not enriched for dipyrimidines. This analysis was mentioned as data not shown in the previous version of the manuscript. Nonetheless, and considering Reviewer 2 comment, the new Figure clarifies the specificity of RNAPII stalling over TT regions and informs on the underlying background signal expected from the difference in RNAPII density along the gene body after wave release.
S8 e	S9 e	New replicates of RNAPII-ser2P ChIP slot-blots include a negative control (NO UV) blot for Input and ChIPs, and were quantified to show the mean FC (\pm SEM) in CPD enrichment between 1h and 2 h of recovery to comply with the the comment of Reviewer 2.
N/A	S10	This new Figure replies to comments from Reviewer 1 and 2 requesting to compare the rate of lesion sensing/repair in DRB-treated and non-treated

		cells. DRB clearly inhibits the specific stalling of RNAPII at TT sites and these results clarify in a quantitative manner the important contribution of de-novo released polymerases in damage recognition.
S9	S11	Numbering change
S10	S12	Numbering change and new analysis  -New panel b indicating the number of samples used in the analysis and the type of sequencing (Whole exome sequencing- WES or Whole Genome Sequencing -WGS) -Old panel b has become c and has been updated to inform on the number of NER specific substitutions used in the analysis. -Old c,e have become d,f and include new analysis for WES data. As the reviewer 2 recommended we have now corrected mutation densities by the underlying exon densities. Analysing separately Lung Adeno WGS data also allowed to validate our methodology for WES data, since both WGS and WES analyses of Lung Adenocarcinoma show comparable distributions profiles. -Old d,f have been updated to e,g accordingly.

Reviewers' Comments:

Reviewer #1:

Remarks to the Author:

Most of reviewer concerns seem to be sufficiently addressed. However, am still confused about how supplemental figure 4 relates to their model. Their model depicts a shift of polymerases from the promoter into the gene or an increase in transcription. However, both panels in figure supplemental figure 4 point to an overall decrease in transcription. Does that mean that polymerases that were already elongating prior to UV treatment somehow stop transcribing or are terminated and released from the chromatin? The latter scenario seems possible since it would agree with the slight decrease in chromatin bound Ser2-P and Ser5-P from the western blots. In short, I believe that their appears to be a wave of Pol II coming from the promoters, but I don't think the authors have fully accounted for what happens to all polymerase in the early time points of these experiments. This is important because termination of already transcribing polymerases would be in agreement with former models and not necessarily challenging them as proposed. In other words, alternative interpretations of the data could satisfy both models, whereby elongating polymerases are terminated and new ones are released from the promoter that take their place and facilitate NER.

This is a lot of high quality work and data, but authors should rethink the interpretations into a model that accounts for all observations made here.

One minor comment: line 233 "Together, our results challenge recent conclusions 49. " I think is appropriate to restate the conclusion that they are challenging for clarity.

Reviewer #2:

Remarks to the Author:

After reading the revised and improved manuscript I find that my concerns were appropriately addressed, and I recommend it for publication.

Reviewer #3:

Remarks to the Author:

With interest I have read the revised version of this manuscript and also went through the rebuttal letter. All my comments have been addressed adequately. I therefore recommend publication of this article without further delay.

We thank all the reviewers for approving our work and recommending our manuscript for publication.

Responses to the Reviewers

Reviewer #1 (Remarks to the Author):

Most of reviewer concerns seem to be sufficiently addressed. However, am still confused about how supplemental figure 4 relates to their model. Their model depicts a shift of polymerases from the promoter into the gene or an increase in transcription. However, both panels in figure supplemental figure 4 point to an overall decrease in transcription. Does that mean that polymerases that were already elongating prior to UV treatment somehow stop transcribing or are terminated and released from the chromatin? The latter scenario seems possible since it would agree with the slight decrease in chromatin bound Ser2-P and Ser5-P from the western blots.

-We thank the Reviewer for pointing this out to us. We apologize that we did not clearly explain in the previous version of our manuscript how the observed changes in nascent RNA levels at the early stages after UV (Supplemental fig 4) relates to our model.

The reviewer is totally right about the fact that theoretically, a number of RNAPII that were elongating prior to UV treatment could stop transcribing or could be terminated and released from the chromatin. Nevertheless, as we had stated in the previous version, our Chip-seq data accounts for a substantial proportion of pri-elongating RNAPII, which remain attached to chromatin in distal regions of gene bodies, as exemplified by the residual binding of RNAPII at TT loci downstream of the wave front (Figure 4 and Supplementary Figure 10). Although, there was a significant reduction of nascent RNA reads in distal regions in comparison to the non-UV conditions, it was clear from our previous Supplementary Figure 4a that the level of nascent RNA in these regions was not zero. We re-analyzed the nascent RNA-seq data to quantify absolute levels of reads (expressed as rpm) and we were able to detect considerable levels of nascent RNA in these regions. In particular, the new Supplementary Figure 4b, shows that there are significantly more nRNA-seq reads in distal regions (50-100 kb downstream of TSS) of active genes longer than 100 kb than in inactive genes. In fact, we also were able to show that analyzing the primary data of *Andrade-Lima et al*, the result we obtained in VH10 cells (hTert wild type human fibroblasts used in our manuscript) was also true for HF1 cells (hTert wild type human fibroblasts used in *Andrade-Lima et al.*). Thus, these data argue against a total loss of the prior to UV elongating RNAPII molecules, which implies that although some RNAPII may stop transcribing or could be terminated and released from the chromatin, there is still a proportion of molecules actively transcribing.

We have re-formulated the text accordingly (page 7, line 177):

“Focusing on pri-elongating RNAPII molecules profiles, we found that there was a substantial retain of RNAPII ChIP-seq reads upon UV in the distal parts of long genes (Fig. 2b, c: compare green and pink). This result was corroborated by the concomitant detection of significant levels of nascent RNA in these regions in active genes (50-100 kb, Supplementary Figure 4b). Notably, meta-analysis of previously reported nascent RNA data⁴⁴ mirrored our findings (Supplementary Figure 4b). Therefore, although elongation rates of pri-elongating complexes were decreased (Fig. 2c, and Supplementary Fig. 5c, d), our data do not support the idea of a total loss of ongoing elongation as suggested previously⁴⁴. We thus conclude that a fraction of RNA polymerases already engaged on gene bodies at the time of stress continues elongating even in distal loci, regardless of the distance to TSS (see below).”

In short, I believe that there appears to be a wave of Pol II coming from the promoters, but I don't think the authors have fully accounted for what happens to all polymerases in the early time points of these experiments. This is important because termination of already transcribing polymerases would be in agreement with former models and not necessarily challenging them as proposed. In other words,

alternative interpretations of the data could satisfy both models, whereby elongating polymerases are terminated and new ones are released from the promoter that take their place and facilitate NER. This is a lot of high quality work and data, but authors should rethink the interpretations into a model that accounts for all observations made here.

-We thank the reviewer for his suggestions in interpreting our data. We have provided alternative interpretations of our conclusions more explicitly in the revised manuscript, page 15, line 432.

“In addition, sending waves of trailing RNAPII molecules throughout the transcribed genome could allow for the next lesions in the gene body to get efficiently detected and repaired even in the case of dissociation from chromatin of the initial damage-detecting and NER-triggering RNAPII molecules^{39,44,57}. In turn, this pathway will ensure a smooth transcription restoration process after repair has been completed regardless of genes’ length and expression levels”.

One minor comment: line 233 "Together, our results challenge recent conclusions 49. " I think is appropriate to restate the conclusion that they are challenging for clarity

This sentence is now completely removed from the revised manuscript. The surrounding paragraph has been modified and is now in p. 8 lines 207-15.